# Sample-Efficient Reinforcement Learning of Undercomplete POMDPs

**Chi Jin**
Princeton University
`chij@princeton.edu`

**Sham M. Kakade**
University of Washington
Microsoft Research, NYC
`sham@cs.washington.edu`

**Akshay Krishnamurthy**
Microsoft Research, NYC
`akshaykr@microsoft.com`

**Qinghua Liu**
Princeton University
`qinghual@princeton.edu`

## Abstract

Partial observability is a common challenge in many reinforcement learning applications, which requires an agent to maintain memory, infer latent states, and integrate this past information into exploration. This challenge leads to a number of computational and statistical hardness results for learning general Partially Observable Markov Decision Processes (POMDPs). This work shows that these hardness barriers do not preclude efficient reinforcement learning for rich and interesting subclasses of POMDPs. In particular, we present a sample-efficient algorithm, *OOM-UCB*, for episodic finite *undercomplete* POMDPs, where the number of observations is larger than the number of latent states and where exploration is essential for learning, thus distinguishing our results from prior works. *OOM-UCB* achieves an optimal sample complexity of $\tilde{\mathcal{O}}(1/\varepsilon^2)$ for finding an $\varepsilon$-optimal policy, along with being polynomial in all other relevant quantities. As an interesting special case, we also provide a computationally and statistically efficient algorithm for POMDPs with deterministic state transitions.

## 1 Introduction

In many sequential decision making settings, the agent lacks complete information about the underlying state of the system, a phenomenon known as *partial observability*. Partial observability significantly complicates the tasks of reinforcement learning and planning, because the non-Markovian nature of the observations forces the agent to maintain memory and reason about beliefs of the system state, all while exploring to collect information about the environment. For example, a robot may not be able to perceive all objects in the environment due to occlusions, and it must reason about how these objects may move to avoid collisions [10]. Similar reasoning problems arise in imperfect information games [8], medical diagnosis [13], and elsewhere [25]. Furthermore, from a theoretical perspective, well-known complexity-theoretic results show that learning and planning in partially observable environments is statistically and computationally intractable in general [23, 22, 30, 21].

The standard formulation for reinforcement learning with partial observability is the *Partially Observable Markov Decision Process* (POMDP), in which an agent operating on noisy observations makes decisions that influence the evolution of a latent state. The complexity barriers apply for this model, but they are of a worst case nature, and they do not preclude efficient algorithms for interesting sub-classes of POMDPs. Thus we ask:

*Can we develop efficient algorithms for reinforcement learning in large classes of POMDPs?*

This question has been studied in recent works [3, 12], which incorporate a decision making component into a long line of work on "spectral methods" for estimation in latent variable models [14, 29, 2, 1], including the Hidden Markov Model. Briefly, these estimation results are based on the method of moments, showing that under certain assumptions the model parameters can be computed by a decomposition of a low-degree moment tensor. The works of Azizzadenesheli et al. [3] and Guo et al. [12] use tensor decompositions in the POMDP setting and obtain sample efficiency guarantees. Neither result considers a setting where strategic exploration is essential for information acquisition, and they do not address one of the central challenges in more general reinforcement learning problems.

**Our contributions.** In this work, we provide new sample-efficient algorithms for reinforcement learning in finite POMDPs in the *undercomplete* regime, where the number of observations is larger than the number of latent states. This assumption is quite standard in the literature on estimation in latent variable models [1]. Our main algorithm *OOM-UCB* uses the principle of optimism for exploration and uses the information gathered to estimate the *Observable Operators* induced by the environment. Our main result proves that *OOM-UCB* finds a near optimal policy for the POMDP using a number of samples that scales polynomially with all relevant parameters and additionally with the minimum singular value of the emission matrix. Notably, *OOM-UCB* finds an $\varepsilon$-optimal policy at the optimal rate of $\tilde{\mathcal{O}}(1/\varepsilon^2)$.

While *OOM-UCB* is statistically efficient for this subclass of POMDPs, we should not expect it to be computationally efficient in general, as this would violate computational barriers for POMDPs. However, in our second contribution, we consider a further restricted subclass of POMDPs in which the latent dynamics are deterministic and where we provide *both* a computationally and statistically efficient algorithm. Notably, deterministic dynamics are still an interesting subclass due to that, while it avoids computational barriers, it still does not mitigate the need for strategic exploration. We prove that our second algorithm has sample complexity scaling with all the relevant parameters as well as the minimum $\ell_2$ distance between emission distributions. This latter quantity replaces the minimum singular value in the guarantee for *OOM-UCB* and is a more favorable dependency.

We provide further motivation for our assumptions with two lower bounds: the first shows that the overcomplete setting is statistically intractable without additional assumptions, while the second necessitates the dependence on the minimum singular value of the emission matrix. In particular, under our assumptions, the agent must engage in strategic exploration for sample-efficiency. As such, the main conceptual advance in our line of inquiry over prior works is that our algorithms address exploration and partial observability in a provably efficient manner.

## 1.1 Related work

A number of computational barriers for POMDPs are known. If the parameters are known, it is PSPACE-complete to compute the optimal policy, and, furthermore, it is NP-hard to compute the optimal memoryless policy [23, 30]. With regards to learning, Mossel and Roch [21] provided an average case computationally complexity result, showing that parameter estimation for a subclass of Hidden Markov Models (HMMs) is at least as hard as learning parity with noise. This directly implies the same hardness result for parameter estimation in POMDP models, due to that an HMM is just a POMDP with a fixed action sequence. On the other hand, for reinforcement learning in POMDPs (in particular, finding a near optimal policy), one may not need to estimate the model, so this lower bound need not directly imply that the RL problem is computational intractable. In this work, we do provide a lower bound showing that reinforcement learning in POMDPs is both statistically and computationally intractable (Propositions 1 and 2).

On the positive side, there is a long history of work on learning POMDPs. [11] studied POMDPs without resets, where the proposed algorithm has sample complexity scaling exponentially with a certain horizon time, which is not possible to relax without further restrictions. [26, 24] proposed to learn POMDPs using Bayesian methods; PAC or regret bounds are not known for these approaches. [18] studied policy gradient methods for learning POMDPs while they considered only Markovian policies and did not address exploration.

Closest to our work are POMDP algorithms based on spectral methods [12, 3], which were originally developed for learning latent variable models [14, 2, 1, 29, 28]. These works give PAC and regret bounds (respectively) for tractable subclasses of POMDPs, but, in contrast with our work, they make

additional assumptions to mitigate the exploration challenge. In [12], it is assumed that all latent states can be reached with nontrivial probability with a constant number of random actions. This allows for estimating the *entire* model without sophisticated exploration. [3] consider a special class of memoryless policies in a setting where all of these policies visit every state and take every action with non-trivial probability. As with [12], this restriction guarantees that the entire model can be estimated regardless of the policy executed, so sophisticated exploration is not required. We also mention that [12, 3] assume that both the transition and observation matrices are full rank, which is stronger than our assumptions. We do not make any assumptions on the transition matrix.

Finally, the idea of representing the probability of a sequence as products of operators dates back to multiplicity automata [27, 9] and reappeared in the Observable Operator Model (OOMs) [16] and Predictive State Representations (PSRs) [20]. While spectral methods have been applied to PSRs [7], we are not aware of results with provable guarantees using this approach. It is also worth mentioning that any POMDP can be modeled as an Input-Output OOM [15].

## 2 Preliminaries

In this section, we define the partially observable Markov decision process, the observable operator model [16], and discuss their relationship.

**Notation.** For any natural number $n \in \mathbb{N}$, we use $[n]$ to denote the set $\{1, 2, \ldots, n\}$. We use bold upper-case letters $\mathbf{B}$ to denote matrices and bold lower-case letters $\mathbf{b}$ to denote vectors. $\mathbf{B}_{ij}$ means the $(i,j)^{\text{th}}$ entry of matrix $\mathbf{B}$ and $(\mathbf{B})_i$ represents its $i^{\text{th}}$ column. For vectors we use $\|\cdot\|_p$ to denote the $\ell_p$-norm, and for matrices we use $\|\cdot\|$, $\|\cdot\|_1$ and $\|\cdot\|_{\text{F}}$ to denote the spectral norm, entrywise $\ell_1$-norm and Frobenius norm respectively. We denote by $\|\mathbf{B}\|_{p \to q} = \max_{\|\mathbf{v}\|_p \leq 1} \|\mathbf{B}\mathbf{v}\|_q$ the $p$-to-$q$ norm of $\mathbf{B}$. For any matrix $\mathbf{B} \in \mathbb{R}^{m \times n}$, we use $\sigma_{\min}(\mathbf{B})$ to denote its smallest singular value, and $\mathbf{B}^\dagger \in \mathbb{R}^{n \times m}$ to denote its Moore-Penrose inverse. For vector $\mathbf{v} \in \mathbb{R}^n$, we denote $\text{diag}(\mathbf{v}) \in \mathbb{R}^{n \times n}$ as a diagonal matrix where $[\text{diag}(\mathbf{v})]_{ii} = \mathbf{v}_i$ for all $i \in [n]$. Finally, we use standard big-O and big-Omega notation $\mathcal{O}(\cdot), \Omega(\cdot)$ to hide only absolute constants which do not depend on any problem parameters, and notation $\tilde{\mathcal{O}}(\cdot), \tilde{\Omega}(\cdot)$ to hide only absolute constants and logarithmic factors.

### 2.1 Partially observable Markov decision processes

We consider an episodic tabular Partially Observable Markov Decision Process (POMDP), which can by specified as POMDP($H, \mathscr{S}, \mathscr{A}, \mathscr{O}, \mathbb{T}, \mathbb{O}, r, \mu_1$). Here $H$ is the number of steps in each episode, $\mathscr{S}$ is the set of states with $|\mathscr{S}| = S$, $\mathscr{A}$ is the set of actions with $|\mathscr{A}| = A$, $\mathscr{O}$ is the set of observations with $|\mathscr{O}| = O$, $\mathbb{T} = \{\mathbb{T}_h\}_{h=1}^H$ specify the transition dynamics such that $\mathbb{T}_h(\cdot|s, a)$ is the distribution over states if action $a$ is taken from state $s$ at step $h \in [H]$, $\mathbb{O} = \{\mathbb{O}_h\}_{h=1}^H$ are emissions such that $\mathbb{O}_h(\cdot|s)$ is the distribution over observations for state $s$ at step $h \in [H]$, $r = \{r_h : \mathscr{O} \to [0,1]\}_{h=1}^H$ are the known deterministic reward functions[1], and $\mu_1(\cdot)$ is the initial distribution over states. Note that we consider nonstationary dynamics, observations, and rewards.

In a POMDP, states are hidden and unobserved to the learning agent. Instead, the agent is only able to see the observations and its own actions. At the beginning of each episode, an initial hidden state $s_1$ is sampled from initial distribution $\mu_1$. At each step $h \in [H]$, the agent first observes $o_h \in \mathscr{O}$ which is generated from the hidden state $s_h \in \mathscr{S}$ according to $\mathbb{O}_h(\cdot|s_h)$, and receives the reward $r_h(o_h)$, which can be computed from the observation $o_h$. Then, the agent picks an action $a_h \in \mathscr{A}$, which causes the environment to transition to hidden state $s_{h+1}$, that is drawn from the distribution $\mathbb{T}_h(\cdot|s_h, a_h)$. The episode ends when $o_H$ is observed.

A policy $\pi$ is a collection of $H$ functions $\left\{\pi_h : \mathscr{T}_h \to \mathscr{A}\right\}_{h \in [H]}$, where $\mathscr{T}_h = (\mathscr{O} \times \mathscr{A})^{h-1} \times \mathscr{O}$ is the set of all possible histories of length $h$. We use $V^\pi \in \mathbb{R}$ to denote the value of policy $\pi$, so that $V^\pi$ gives the expected cumulative reward received under policy $\pi$:

$$V^\pi := \mathbb{E}_\pi \left[ \sum_{h=1}^H r_h(o_h) \right].$$

Since the state, action, observation spaces, and the horizon, are all finite, there always exists an optimal policy $\pi^\star$ which gives the optimal value $V^\star = \sup_\pi V^\pi$. We remark that, in general, the optimal policy of a POMDP will select actions based the entire history, rather than just the recent observations and actions. This is one of the major differences between POMDPs and standard Markov Decision Processes (MDPs), where the optimal policies are functions of the most recently observed state. This difference makes POMDPs significantly more challenging to solve.

**The POMDP learning objective.** Our objective in this paper is to learn an $\varepsilon$-**optimal policy** $\hat{\pi}$ in the sense that $V^{\hat{\pi}} \geq V^\star - \varepsilon$, using a polynomial number of samples.

## 2.2 The observable operator model

We have described the POMDP model via the transition and observation distributions $\mathbb{T}, \mathbb{O}$ and the initial distribution $\mu_1$. While this parametrization is natural for describing the dynamics of the system, POMDPs can also be fully specified via a different set of parameters: a set of operators $\{\mathbf{B}_h(a, o) \in \mathbb{R}^{O \times O}\}_{h \in [H-1], a \in \mathscr{A}, o \in \mathscr{O}}$, and a vector $\mathbf{b}_0 \in \mathbb{R}^O$.

In the undercomplete setting where $S \leq O$ and where observation probability matrices $\{\mathbb{O}_h \in \mathbb{R}^{O \times S}\}_{h \in [H]}$ are all full column-rank, the operators $\{\mathbf{B}_h(a, o)\}_{h,a,o}$ and vector $\mathbf{b}_0$ can be expressed in terms of $(\mathbb{T}, \mathbb{O}, \mu_1)$ as follows:

$$\mathbf{B}_h(a, o) = \mathbb{O}_{h+1} \mathbb{T}_h(a) \operatorname{diag}(\mathbb{O}_h(o|\cdot)) \mathbb{O}_h^\dagger, \qquad \mathbf{b}_0 = \mathbb{O}_1 \mu_1. \tag{1}$$

where we use the matrix and vector notation for $\mathbb{O}_h \in \mathbb{R}^{O \times S}$ and $\mu_1 \in \mathbb{R}^S$ here, such that $[\mathbb{O}_h]_{o,s} = \mathbb{O}_h(o|s)$ and $[\mu_1]_s = \mu_1(s)$. $\mathbb{T}_h(a) \in \mathbb{R}^{S \times S}$ denotes the transition matrix given action $a \in \mathscr{A}$ where $[\mathbb{T}_h(a)]_{s',s} = \mathbb{T}_h(s'|s, a)$, and $\mathbb{O}_h(o|\cdot) \in \mathbb{R}^S$ denotes the $o$-th row in matrix $\mathbb{O}_h$ with $[\mathbb{O}_h(o|\cdot)]_s = \mathbb{O}_h(o|s)$. Note that the matices defined in (1) have rank at most $S$. Using these matrices $\mathbf{B}_h$, it can be shown that (Appendix E.1), for any sequence of $(o_H, \ldots, a_1, o_1) \in \mathscr{O} \times (\mathscr{A} \times \mathscr{O})^{H-1}$, we have:

$$\mathbb{P}(o_H, \ldots, o_1 | a_{H-1}, \ldots, a_1) = \mathbf{e}_{o_H}^\top \cdot \mathbf{B}_{H-1}(a_{H-1}, o_{H-1}) \cdots \mathbf{B}_1(a_1, o_1) \cdot \mathbf{b}_0. \tag{2}$$

Describing these conditional probabilities for every sequence is sufficient to fully specify the entire dynamical system. Therefore, as an alternative to directly learning $\mathbb{T}, \mathbb{O}$ and $\mu_1$, it is also sufficient to learn operators $\{\mathbf{B}_h(a, o)\}_{h,a,o}$ and vector $\mathbf{b}_0$ in order to learn the optimal policy. The latter approach enjoys the advantage that (2) does not explicitly involve latent variables. It refers only to observable quantities—actions and observations.

We remark that the operator model introduced in this section (which is parameterized by $\{\mathbf{B}_h(a, o)\}_{h,a,o}$ and $\mathbf{b}_0$) bears significant similarity to Jaeger's Input-Output Observable Operator Model (IO-OOM) [16], except a few minor technical differences.[2] With some abuse of terminology, we also refer to our model as Observable Operator Model (OOM) in this paper. It is worth noting that Jaeger's IO-OOMs are strictly more general than POMDPs [16] and also includes overcomplete POMDPs via a relation different from (1). Since our focus is on undercomplete POMDPs, we refer the reader to [16] for more details.

# 3 Main Results

We first state our main assumptions, which we motivate with corresponding hardness results in their absence. We then present our main algorithm, *OOM-UCB*, along with its sample efficiency guarantee.

## 3.1 Assumptions

In this paper, we make the following assumptions.

**Assumption 1.** We assume the POMDP is undercomplete, i.e. $S \leq O$. We also assume the minimum singular value of the observation probability matrices $\sigma_{\min}(\mathbb{O}_h) \geq \alpha > 0$ for all $h \in [H]$.

Both assumptions are standard in the literature on learning Hidden Markov Models (HMMs)—an uncontrolled version of POMDP [see e.g., 2]. The second assumption that $\sigma_{\min}(\mathbb{O}_h)$ is lower-bounded

**Algorithm 1** Observable Operator Model with Upper Confidence Bound (OOM-UCB)

---

1: **Initialize:** set all entries in a vector of counts $\mathbf{n} \in \mathbb{N}^O$, and in matrices of counts $\mathbf{N}_h(a, \tilde{a}) \in \mathbb{N}^{O \times O}$, $\mathbf{M}_h(o, a, \tilde{a}) \in \mathbb{N}^{O \times O}$ to be zero for all $(o, a, \tilde{a}) \in \mathscr{O} \times \mathscr{A}^2$

2: set confidence set $\Theta_1 \leftarrow \cap_{h \in [H]} \{\hat{\theta} \mid \sigma_{\min}(\hat{\mathbb{O}}_h) \geq \alpha\}$.

3: **for** $k = 1, 2, \ldots, K$ **do**

4:   compute the optimistic policy $\pi_k \leftarrow \mathrm{argmax}_\pi \max_{\hat{\theta} \in \Theta_k} V^\pi(\hat{\theta})$.

5:   observe $o_1$, and set $\mathbf{n} \leftarrow \mathbf{n} + \mathbf{e}_{o_1}$

6:   $\mathfrak{b} \leftarrow (\cap_{h \in [H]} \{\hat{\theta} \mid \sigma_{\min}(\hat{\mathbb{O}}_h) \geq \alpha\}) \cap \{\hat{\theta} \mid \|k \cdot \mathbf{b}_0(\hat{\theta}) - \mathbf{n}\|_2 \leq \beta_k\}$.

7:   **for** $(h, a, \tilde{a}) \in [H-1] \times \mathscr{A}^2$ **do**

8:     execute policy $\pi_k$ from step 1 to step $h - 2$.

9:     take action $\tilde{a}$ at step $h - 1$, and action $a$ at step $h$ respectively.

10:     observe $(o_{h-1}, o_h, o_{h+1})$, and set $\mathbf{N}_h(a, \tilde{a}) \leftarrow \mathbf{N}_h(a, \tilde{a}) + \mathbf{e}_{o_h} \mathbf{e}_{o_{h-1}}^\top$.

11:     set $\mathbf{M}_h(o_h, a, \tilde{a}) \leftarrow \mathbf{M}_h(o_h, a, \tilde{a}) + \mathbf{e}_{o_{h+1}} \mathbf{e}_{o_{h-1}}^\top$.

12:     $\mathfrak{B}_h(a, \tilde{a}) \leftarrow \cap_{o \in \mathscr{O}} \{\hat{\theta} \mid \|\mathbf{B}_h(a, o; \hat{\theta})\mathbf{N}_h(a, \tilde{a}) - \mathbf{M}_h(o, a, \tilde{a})\|_F \leq \gamma_k\}$.

13:   construct the confidence set $\Theta_{k+1} \leftarrow [\cap_{(h, a, \tilde{a}) \in [H-1] \times \mathscr{A}^2} \mathfrak{B}_h(a, \tilde{a})] \cap \mathfrak{b}$.

14: **Output:** $\pi_k$ where $k$ is sampled uniformly from $[K]$.

---

is a robust version of the assumption that $\mathbb{O}_h \in \mathbb{R}^{O \times S}$ is full column-rank, which is equivalent to $\sigma_{\min}(\mathbb{O}_h) > 0$. Together, these assumption ensure that the observations will contain a reasonable amount of information about the latent states.

We do not assume that the initial distribution $\mu_1$ has full support, nor do we assume the transition probability matrices $\mathbb{T}_h$ are full rank. In fact, Assumption 1 is *not* sufficient for identification of the system, i.e. recovering parameters $\mathbb{T}, \mathbb{O}, \mu_1$ in total-variance distance. Exploration is crucial to find a near-optimal policy in our setting.

We motivate both assumptions above by showing that, with absence of either one, learning a POMDP is statistically intractable. That is, it would require an exponential number of samples for any algorithm to learn a near-optimal policy with constant probability.

**Proposition 1.** *For any algorithm $\mathfrak{A}$, there exists an overcomplete POMDP ($S > O$) with $S$ and $O$ being small constants, which satisfies $\sigma_{\min}(\mathbb{O}_h) = 1$ for all $h \in [H]$, such that algorithm $\mathfrak{A}$ requires at least $\Omega(A^{H-1})$ samples to ensure learning a $(1/4)$-optimal policy with probability at least $1/2$.*

**Proposition 2.** *For any algorithm $\mathfrak{A}$, there exists an undercomplete POMDP ($S \leq O$) with $S$ and $O$ being small constants, such that algorithm $\mathfrak{A}$ requires at least $\Omega(A^{H-1})$ samples to ensure learning a $(1/4)$-optimal policy with probability at least $1/2$.*

Proposition 1 and 2 are both proved by constructing hard instances, which are modifications of classical combinatorial locks for MDPs [19]. We refer readers to Appendix B for more details.

### 3.2   Algorithm

We are now ready to describe our algorithm. Assumption 1 enables the representation of the POMDP using OOM with relation specified as in Equation (1). Our algorithm, Observable Operator Model with Upper Confidence Bound (OOM-UCB, algorithm 1), is an optimistic algorithm which heavily exploits the OOM representation to obtain valid uncertainty estimates of the parameters of the underlying model.

To condense notation in Algorithm 1, we denote the parameters of a POMDP as $\theta = (\mathbb{T}, \mathbb{O}, \mu_1)$. We denote $V^\pi(\theta)$ as the value of policy $\pi$ if the underlying POMDP has parameter $\theta$. We also write the parameters of the OOM $(\mathbf{b}_0(\theta), \mathbf{B}_h(a, o; \theta))$ as a function of parameter $\theta$, where the dependency is specified as in (1). We adopt the convention that at the 0-th step, the observation $o_0$ and state $s_0$ are always set to be some fixed dummy observation and state, and, starting from $s_0$, the environment transitions to $s_1$ with distribution $\mu_1$ regardless of what action $a_0$ is taken.

At a high level, Algorithm 1 is an iterative algorithm that, in each iteration, (a) computes an optimistic policy and model by maximizing the value (Line 4) subject to a given confidence set constraint, (b) collects data using the optimistic policy, and (c) incorporates the data into an updated confidence

set for the OOM parameters (Line 5-13). The first two parts are straightforward, so we focus the discussion on computing the confidence set. We remark that in general the optimization in Line 4 may not be solved in polynomial time (see discussions of the computational complexity after Theorem 3).

First, since $\mathbf{b}_0$ in (1) is simply the probability over observations at the first step, our confidence set for $\mathbf{b}_0$ in Line 6 is simply based on counting the number of times each observation appears in the first step and Hoeffding's concentration inequality.

Our construction of the confidence sets for the operators $\{\mathbf{B}_h(a, o)\}_{h,a,o}$ is inspired by the method-of-moments estimator in HMM literature [14]. Consider two fixed actions $a, \tilde{a}$, and an arbitrary distribution over $s_{h-1}$. Let $\mathbf{P}_h(a, \tilde{a}), \mathbf{Q}_h(o, a, \tilde{a}) \in \mathbb{R}^{O \times O}$ be the probability matrices such that

$$[\mathbf{P}_h(a, \tilde{a})]_{o',o''} = \mathbb{P}(o_h = o', o_{h-1} = o'' | a_h = a, a_{h-1} = \tilde{a}),$$
$$[\mathbf{Q}_h(o, a, \tilde{a})]_{o',o''} = \mathbb{P}(o_{h+1} = o', o_h = o, o_{h-1} = o'' | a_h = a, a_{h-1} = \tilde{a}). \tag{3}$$

It can be verified that $\mathbf{B}_h(a, o)\mathbf{P}_h(a, \tilde{a}) = \mathbf{Q}_h(o, a, \tilde{a})$ (Fact 17 in the appendix). Our confidence set construction (Line 12 in Algorithm 1) is based on this fact: we replace the probability matrices $\mathbf{P}, \mathbf{Q}$ by empirical estimates $\mathbf{N}, \mathbf{M}$, and we use concentration inequalities to determine the width of the confidence set. Finally, our overall confidence set for the parameters $\theta$ is simply the intersection of the confidence sets for all induced operators and $\mathbf{b}_0$, additionally incorporating the constraint on $\sigma_{\min}(\mathbb{O}_h)$ from Assumption 1.

### 3.3 Theoretical guarantees

Our OOM-UCB algorithm enjoys the following sample complexity guarantee.

**Theorem 3.** *For any $\varepsilon \in (0, H]$, there exists $K_{\max} = \mathrm{poly}(H, S, A, O, \alpha^{-1})/\varepsilon^2$ and an absolute constant $c_1$, such that for any POMDP that satisfies Assumption 1, if we set hyperparameters $\beta_k = c_1\sqrt{k \log(KAOH)}$, $\gamma_k = \sqrt{S}\beta_k/\alpha$, and $K \geq K_{\max}$, then the output policy $\hat{\pi}$ of Algorithm 1 will be $\varepsilon$-optimal with probability at least $2/3$.*

Theorem 3 claims that in polynomially many iterations of the outer loop, Algorithm 1 learns a near-optimal policy for any undercomplete POMDP that satisfies Assumption 1. Since our algorithm only uses $O(H^2 A^2)$ samples per iteration of the outer loop, this implies that the sample complexity is also $\mathrm{poly}(H, S, A, O, \alpha^{-1})/\varepsilon^2$. We remark that the $1/\varepsilon^2$ dependence is optimal, which follows from standard concentration arguments. To the best of our knowledge, this is the first sample efficiency result for learning a class of POMDPs where exploration is essential. [3]

While Theorem 3 does guarantee sample efficiency, Algorithm 1 is not computationally efficient due to that the computation of the optimistic policy (Line 4) may not admit a polynomial time implementation, which should be expected given the aforementioned computational complexity results. We now turn to a further restricted (and interesting) subclass of POMDPs where we can address *both* the computational and statistical challenges.

## 4 Results for POMDPs with Deterministic Transition

In this section, we complement our main result by investigating the class of POMDPs with deterministic transitions, where both computational and statistical efficiency can be achieved. We say a POMDP is of *deterministic transition* if both its transition and initial distribution are deterministic, i.e, if the entries of matrices $\{\mathbb{T}_h\}_h$ and vector $\mu_1$ are either 0 or 1. We remark that while deterministic dynamics avoids computational barriers, it does not mitigate the need for exploration.

Instead of Assumption 1, for the deterministic transition case, we require that the columns of the observation matrices $\mathbb{O}_h$ are well-separated.

**Assumption 2.** For any $h \in [H]$, $\min_{s \neq s'} \|\mathbb{O}_h(\cdot|s) - \mathbb{O}_h(\cdot|s')\| \geq \xi$.

Assumption 2 guarantees that observation distributions for different states are sufficiently different, by at least $\xi$ in Euclidean norm. It does not require that the POMDP is undercomplete, and, in fact, is strictly weaker than Assumption 1. In particular, for undercomplete models,

$\min_{s \neq s'} \|\mathbb{O}_h(\cdot|s) - \mathbb{O}_h(\cdot|s')\| \geq \sqrt{2}\sigma_{\min}(\mathbb{O}_h)$, and so Assumption 1 implies Assumption 2 for $\xi = \sqrt{2}\alpha$.

Leveraging deterministic transitions, we can design a specialized algorithm (Algorithm 2 in the appendix) that learns an $\varepsilon$-optimal policy using polynomially many samples and in polynomial time. We present the formal theorem here, and refer readers to Appendix D for more details.

**Theorem 4.** *For any $p \in (0, 1]$, there exists an algorithm such that for any deterministic transition POMDP satisfying Assumption 2, within $\mathcal{O}\left(H^2 SA \log(HSA/p)/(\min\{\varepsilon/(\sqrt{O}H), \xi\})^2\right)$ samples and computations, the output policy of the algorithm is $\varepsilon$-optimal with probability at least $1 - p$.*

## 5 Analysis Overview

In this section, we provide an overview of the proof of our main result—Theorem 3. Please refer to Appendix C for the full proof.

We start our analysis by noticing that the output policy $\hat{\pi}$ of Algorithm 1 is uniformly sampled from $\{\pi_k\}_{k=1}^K$ computed in the algorithm. If we can show that

$$(1/K) \sum_{k=1}^K V^\star - V^{\pi_k} \leq \varepsilon/10, \qquad (4)$$

then at least a $2/3$ fraction of the policies in $\{\pi_k\}_{k=1}^K$ must be $\varepsilon$-optimal, and uniform sampling would find such a policy with probability at least $2/3$. Therefore, our proof focuses on achieving (4).

We begin by conditioning on the event that for each iteration $k$, our constructed confidence set $\Theta_k$ in fact contains the true parameters $\theta^\star = (\mathbb{T}, \mathbb{O}, \mu_1)$ of the POMDP. This holds with high probability and is achieved by setting the widths $\beta_k$ and $\gamma_k$ appropriately (see Lemma 14 in the appendix).

### 5.1 Bounding suboptimality in value by error in density estimation

Line 4 of Algorithm 1 computes the greedy policy $\pi_k \leftarrow \operatorname{argmax}_\pi \max_{\hat{\theta} \in \Theta_k} V^\pi(\hat{\theta})$ with respect to the current confidence set $\Theta_k$. Let $\theta_k$ denote the maximizing model parameters in the $k$-th iteration. As $(\pi_k, \theta_k)$ are optimistic, we have $V^\star \equiv V^\star(\theta^\star) \leq V^{\pi_k}(\theta_k)$ for all $k \in [K]$. Thus, for any $k \in [K]$:

$$V^\star - V^{\pi_k} \leq V^{\pi_k}(\theta_k) - V^{\pi_k}(\theta^\star) \leq H \sum_{o_H,\ldots,o_1} |\mathbb{P}_{\theta_k}^{\pi_k}(o_H,\ldots,o_1) - \mathbb{P}_{\theta^\star}^{\pi_k}(o_H,\ldots,o_1)|, \quad (5)$$

where $\mathbb{P}_\theta^\pi$ denotes the probability measure over observations under policy $\pi$ for POMDP with parameters $\theta$. The second inequality holds because the cumulative reward is a function of observations $(o_H,\ldots,o_1)$ and is upper bounded by $H$. This upper bounds the suboptimality in value by the total variation distance between the $H$-step observation distributions.

Next, note that we can always choose the greedy policy $\pi_k$ to be deterministic, i.e., the probability to take any action given a history is either $0$ or $1$. This allows us to define the following set for any deterministic policy $\pi$:

$$\Gamma(\pi, H) := \{\tau_H = (o_H,\ldots,a_1,o_1) \mid \pi(a_{H-1},\ldots,a_1|o_H,\ldots,o_1) = 1\}.$$

In words, $\Gamma(\pi, H)$ is a set of all the observation and action sequences of length $H$ that could occur under the $\pi$. For any deterministic policy $\pi$, there is a one-to-one correspondence between $\mathscr{O}^H$ and $\Gamma(\pi, H)$ and moreover, for any sequence $\tau_H = (o_H,\ldots,a_1,o_1) \in \Gamma(\pi, H)$, we have:

$$p(\tau_H; \theta) := \mathbb{P}_\theta(o_H,\ldots,o_1|a_{H-1},\ldots,a_1) = \mathbb{P}_\theta^\pi(o_H,\ldots,o_1). \qquad (6)$$

The derivation of equation (6) can be found in Appendix E.2. Combining this with (5) and summing over all episodes, we conclude that:

$$\sum_{k=1}^K (V^\star - V^{\pi_k}) \leq H \sum_{k=1}^K \sum_{\tau_H \in \Gamma(\pi_k, H)} |p(\tau_H; \theta_k) - p(\tau_H; \theta^\star)|.$$

This upper bounds the suboptimality in value by errors in estimating the conditional probabilities.

## 5.2 Bounding error in density estimation by error in estimating operators

For the next step, we leverage the OOM representation to bound the difference between the conditional probabilities $p(\tau_H; \theta_k)$ and $p(\tau_H; \theta^\star)$. Recall that from (2), the conditional probability can be written as a product of the observable operators for each step and $\mathbf{b}_0$. Therefore, for any two parameters $\hat{\theta}$ and $\theta$, we have following relation for any sequence $\tau_H = (o_H, \ldots, a_1, o_1)$:

$$p(\tau_H; \hat{\theta}) - p(\tau_H; \theta) = \mathbf{e}_{o_H}^\top \cdot \mathbf{B}_{H-1}(a_{H-1}, o_{H-1}; \hat{\theta}) \cdots \mathbf{B}_1(a_1, o_1; \hat{\theta}) \cdot [\mathbf{b}_0(\hat{\theta}) - \mathbf{b}_0(\theta)]$$

$$+ \sum_{h=1}^{H-1} \mathbf{e}_{o_H}^\top \cdot \mathbf{B}_{H-1}(a_{H-1}, o_{H-1}; \hat{\theta}) \cdots [\mathbf{B}_h(a_h, o_h; \hat{\theta}) - \mathbf{B}_h(a_h, o_h; \theta)] \cdots \mathbf{B}_1(a_1, o_1; \theta) \cdot \mathbf{b}_0(\theta).$$

This relates the difference $p(\tau_H; \hat{\theta}) - p(\tau_H; \theta)$ to the differences in operators and $\mathbf{b}_0$. Formally, with further relaxation and summation over all sequence in $\Gamma(\pi, H)$, we have the following lemma (also see Lemma 10 in Appendix C).

**Lemma 5.** *Given a deterministic policy $\pi$ and two sets of undercomplete POMDP parameters $\theta = (\mathbb{O}, \mathbb{T}, \mu_1)$ and $\hat{\theta} = (\hat{\mathbb{O}}, \hat{\mathbb{T}}, \hat{\mu}_1)$ with $\sigma_{\min}(\hat{\mathbb{O}}) \geq \alpha$, we have*

$$\sum_{\tau_H \in \Gamma(\pi, H)} |p(\tau_H; \hat{\theta}) - p(\tau_H; \theta)| \leq \frac{\sqrt{S}}{\alpha} \left( \|\mathbf{b}_0(\hat{\theta}) - \mathbf{b}_0(\theta)\|_1 + \sum_{(a,o) \in \mathscr{A} \times \mathscr{O}} \|[\mathbf{B}_1(a, o; \hat{\theta}) - \mathbf{B}_1(a, o; \theta)]\mathbf{b}_0(\theta)\|_1 \right.$$

$$\left. + \sum_{h=2}^{H-1} \sum_{(a,\tilde{a},o) \in \mathscr{A}^2 \times \mathscr{O}} \sum_{s=1}^{S} \left\| \left( \mathbf{B}_h(a, o; \hat{\theta}) - \mathbf{B}_h(a, o; \theta) \right) \mathbb{O}_h \mathbb{T}_{h-1}(\tilde{a}) \mathbf{e}_s \right\|_1 \mathbb{P}_\theta^\pi(s_{h-1} = s) \right). \quad (7)$$

This lemma suggests that if we could estimate the operators accurately, we would have small value sub-optimality. However, Assumption 1 is not sufficient for parameter recovery. It is possible that in some step $h$, there exists a state $s_h$ that can be reached with only very small probability no matter what policy is used. Since we cannot collect many samples from $s_h$, it is not possible to estimate the corresponding component in the operator $B_h$. In other words, we cannot hope to make $\|\mathbf{B}_h(a, o; \hat{\theta}) - \mathbf{B}_h(a, o; \theta)\|_1$ small in our setting.

To proceed, it is crucial to observe that the third term on the RHS of (7), is in fact the operator error $\mathbf{B}_h(a, o; \hat{\theta}) - \mathbf{B}_h(a, o; \theta)$ projected onto the direction $\mathbb{O}_h \mathbb{T}_{h-1}(\tilde{a}) \mathbf{e}_s$ and additionally reweighted by the probability of visiting state $s$ in step $h-1$. Therefore, if $s$ is hard to reach, the weighting probability will be very small, which means that even though we cannot estimate $\mathbf{B}_h(a, o; \theta)$ accurately in the corresponding direction, it has a negligible contribution to the density estimation error (LHS of (7)).

## 5.3 Bounding error in estimating operators by OOM-UCB algorithm

By Lemma 5, we only need to bound the error in operators reweighted by visitation probability. This is achieved by a careful design of the confidence sets in the OOM-UCB algorithm. This construction is based on the method of moments, which heavily exploits the undercompleteness of the POMDP. To showcase the main idea, we focus on bounding the third term on the RHS of (7).

Consider a fixed $(o, a, \tilde{a})$ tuple, a fixed step $h \in [H]$, and a fixed iteration $k \in [K]$. We define moment matrices $\mathbf{P}_h(a, \tilde{a}), \mathbf{Q}_h(o, a, \tilde{a}) \in \mathbb{R}^{O \times O}$ as in (3) for distribution on $s_{h-1}$ that equals $(1/k) \cdot \sum_{t=1}^{k} \mathbb{P}_{\theta^\star}^{\pi_t}(s_{h-1} = \cdot)$. We also denote $\hat{\mathbf{P}}_h(a, \tilde{a}) = \mathbf{N}_h(a, \tilde{a})/k, \hat{\mathbf{Q}}_h(o, a, \tilde{a}) = \mathbf{M}_h(o, a, \tilde{a})/k$ for $\mathbf{N}_h, \mathbf{M}_h$ matrices *after the update* in the $k$-th iteration of Algorithm 1. By martingale concentration, it is not hard to show that with high probability:

$$\|\mathbf{P}_h(a, \tilde{a}) - \hat{\mathbf{P}}_h(a, \tilde{a})\|_F \leq \tilde{\mathcal{O}}(1/\sqrt{k}), \qquad \|\mathbf{Q}_h(o, a, \tilde{a}) - \hat{\mathbf{Q}}_h(o, a, \tilde{a})\|_F \leq \tilde{\mathcal{O}}(1/\sqrt{k}).$$

Additionally, we can show that for the true operator and the true moments, we have $\mathbf{B}_h(a, o; \theta^\star)\mathbf{P}_h(a, \tilde{a}) = \mathbf{Q}_h(o, a, \tilde{a})$. Meanwhile, by the construction of our confidence set $\Theta_{k+1}$, we know that for any $\hat{\theta} \in \Theta_{k+1}$, we have

$$\|\mathbf{B}_h(a, o; \hat{\theta})\hat{\mathbf{P}}_h(a, \tilde{a}) - \hat{\mathbf{Q}}_h(o, a, \tilde{a})\|_F \leq \gamma_k/k.$$

Combining all relations above, we see that $\mathbf{B}_h(a, o; \hat{\theta})$ is accurate in the directions spanned by $\mathbf{P}_h(a, \tilde{a})$, which, by definition, are directions frequently visited by the previous policies $\{\pi_t\}_{t=1}^{k}$.

Formally, we have the following lemma, which allows us to bound the third term on the RHS of (7) using the algebraic transformation in Lemma 16.

**Lemma 6.** *With probability at least $1 - \delta$, for all $k \in [K]$, for any $\hat{\theta} = (\hat{\mathbb{O}}, \hat{\mathbb{T}}, \hat{\mu}_1) \in \Theta_{k+1}$ and $(o, a, \tilde{a}, h) \in \mathscr{O} \times \mathscr{A}^2 \times \{2, \ldots, H-1\}$, and $\iota = \log(KAOH/\delta)$, we have*

$$\sum_{s=1}^{S} \left\| \left( \mathbf{B}_h(a, o; \hat{\theta}) - \mathbf{B}_h(a, o; \theta^\star) \right) \mathbb{O}_h \mathbb{T}_{h-1}(\tilde{a}) \mathbf{e}_s \right\|_1 \sum_{t=1}^{k} \mathbb{P}_{\theta^\star}^{\pi_t}(s_{h-1} = s) \leq \mathcal{O}\left( \sqrt{\frac{kS^2O\iota}{\alpha^4}} \right).$$

## 6 Conclusion

In this paper, we give a sample efficient algorithm for reinforcement learning in undercomplete POMDPs. Our results leverage a connection to the observable operator model and employ a refined error analysis. To our knowledge, this gives the first provably efficient algorithm for strategic exploration in partially observable environments.

## Broader Impact

As this is a theoretical contribution, we do not envision that our direct results will have a tangible societal impact. Our broader line of inquiry could impact a line of thinking in a way that provides additional means to provide confidence intervals relevant for planning and learning. There is an increasing needs for applications to understand planning under uncertainty in the broader context of safety and reliability, and POMDPs provide one potential framework.

## Funding Disclosure

This work was supported by Microsoft and Princeton University. S.K. gratefully acknowledges funding from the ONR award N00014-18-1-2247, NSF Awards CCF-1703574 and CCF-1740551.

## Footnotes

[1]Since rewards are observable in most applications, it is natural to assume the reward is a known function of the observation. While we study deterministic reward functions for notational simplicity, our results generalize to randomized reward functions. Also, we assume the reward is in $[0, 1]$ without loss of generality.

[2]Jaeger's IO-OOM further requires the column-sums of operators to be 1.

[3]See Appendix C.4 for the explicit polynomial dependence of sample complexity; here, the success probability is a constant, but one can make it arbitrarily close to 1 by a standard boosting trick (see Appendix E.3 ).

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
