[Supplementary Material]

# A Notation

Below, we introduce some notations that will be used in appendices.

| notation | definition |
|----------|------------|
| $\mathbf{n}^k$ | value of $\mathbf{n}$ *after the update* in the $k^{\text{th}}$ iteration of Algorithm 1 |
| $\mathbf{N}_h^k(a, \tilde{a})$ | value of $\mathbf{N}_h(a, \tilde{a})$ *after the update* in the $k^{\text{th}}$ iteration of Algorithm 1 |
| $\mathbf{M}_h^k(o, a, \tilde{a})$ | value of $\mathbf{M}_h(o, a, \tilde{a})$ *after the update* in the $k^{\text{th}}$ iteration of Algorithm 1 |
| $\theta$ | a parameter triple $(\mathbb{T}, \mathbb{O}, \mu_1)$ of a POMDP |
| $\theta^\star$ | the groundtruth POMDP parameter triple |
| POMDP$(\theta)$ | POMDP$(H, \mathscr{S}, \mathscr{A}, \mathscr{O}, \mathbb{T}, \mathbb{O}, r, \mu_1)$ |
| $\tau_h$[4] | a length-$h$ trajectory: $\tau_h = [a_h, o_h, \ldots, a_1, o_1] \in (\mathscr{A} \times \mathscr{O})^h$ |
| $\Gamma(\pi, h)$[5] | $\{\tau_h = (a_h, o_h, \ldots, a_1, o_1) \mid \pi(a_h, \ldots, a_1\|o_h, \ldots, o_1) = 1\}$. |
| $\mathbf{b}(\tau_h; \theta)$ | $\mathbf{B}_h(a_h, o_h; \theta) \cdots \mathbf{B}_1(a_1, o_1; \theta) \cdot \mathbf{b}_0(\theta)$ |
| $\mathbb{P}_\theta^\pi(s_h = s)$ | probability of visiting state $s$ at $h^{\text{th}}$ step when executing policy $\pi$ on POMDP$(\theta)$ |
| $\mathbf{1}(x = y)$ | equal to 1 if $x = y$ and 0 otherwise. |
| $\mathbf{e}_o$ | an $O$-dimensional vector with $(\mathbf{e}_o)_i = \mathbf{1}(o = i)$ |
| $(\mathbf{X})_o$ | the $o^{\text{th}}$ column of matrix $\mathbf{X}$ |
| $\mathbf{I}_n$ | $n \times n$ identity matrix |
| $C_{\text{poly}}$ | $\text{poly}(S, O, A, H, 1/\alpha, \log(1/\delta))$ |
| $\iota$ | $\log(AOHK/\delta)$ |

Let $\mathbf{x} \in \mathbb{R}^{n_x}$, $\mathbf{y} \in \mathbb{R}^{n_y}$ and $\mathbf{z} \in \mathbb{R}^{n_z}$. We denote by $\mathbf{x} \otimes \mathbf{y} \otimes \mathbf{z}$ the tensor product of vectors $\mathbf{x}$, $\mathbf{y}$ and $\mathbf{z}$, an $n_x \times n_y \times n_z$ tensor with $(i, j, k)^{\text{th}}$ entry equal to $\mathbf{x}_i \mathbf{y}_j \mathbf{z}_k$. Let $\mathbf{X} \in \mathbb{R}^{n_X \times m}$, $\mathbf{Y} \in \mathbb{R}^{n_Y \times m}$ and $\mathbf{Z} \in \mathbb{R}^{n_Z \times m}$. We generalize the notation of tensor product to matrices by defining $\mathbf{X} \otimes \mathbf{Y} \otimes \mathbf{Z} = \sum_{l=1}^m (\mathbf{X})_l \otimes (\mathbf{Y})_l \otimes (\mathbf{Z})_l$, which is an $n_X \times n_Y \times n_Z$ tensor with $(i, j, k)^{\text{th}}$ entry equal to $\sum_{l=1}^m \mathbf{X}_{il} \mathbf{Y}_{jl} \mathbf{Z}_{kl}$.

Let $X$ be a random variable taking value in $[m]$, we denote by $\mathbb{P}(X = \cdot)$ an $m$-dimensional vector whose $i^{\text{th}}$ entry is $\mathbb{P}(X = i)$.

# B  Proof of Hardness Results

The hard examples constructed below are variants of the ones used in [19].

**Proposition 1.** *For any algorithm $\mathfrak{A}$, there exists an overcomplete POMDP ($S > O$) with $S$ and $O$ being small constants, which satisfies $\sigma_{\min}(\mathbb{O}_h) = 1$ for all $h \in [H]$, such that algorithm $\mathfrak{A}$ requires at least $\Omega(A^{H-1})$ samples to ensure learning a $(1/4)$-optimal policy with probability at least $1/2$.*

*Proof.* Consider the following $H$-step nonstationary POMDP:

1. STATE There are four states: two good states $g_1$ and $g_2$ and two bad states $b_1$ and $b_2$. The initial state is picked uniformly at random.

2. OBSERVATION There are only two different observations $u_1$ and $u_2$. At step $h \in [H-1]$, we always observe $u_1$ at $g_1$ and $b_1$, and observe $u_2$ at $g_2$ and $b_2$. At step $H$, we always observe $u_1$ at good states and $u_2$ at bad states. It's direct to verify $\sigma_{\min}(\mathbb{O}_h) = 1$ for all $h \in [H]$.

3. REWARD There is no reward at the fist $H-1$ steps (i.e. $r_h = 0$ for all $h \in [H-1]$). At step $H$, we receive reward 1 if we observe $u_1$ and no reward otherwise (i.e. $r_H(o) = \mathbf{1}(o = u_1)$).

4. TRANSITION There is one good action $a_h^\star$ and $A - 1$ bad actions for each $h \in [H-1]$. At step $h \in [H-1]$, suppose we are at a good state ($g_1$ or $g_2$), then we will transfer to $g_1$ or $g_2$ uniformly at random if we take $a_h^\star$ and otherwise transfer to $b_1$ or $b_2$ uniformly at random. In contrast, if we are at a bad state ($b_1$ or $b_2$), we will always transfer to $b_1$ or $b_2$ uniformly at random no matter what action we take. Note that two good (bad) states are equivalent in terms of transition.

We have the following key observations:

1. Once we are at bad states, we always stay at bad states.

2. We have

$$\mathbb{P}(o_{1:H-1} = z \mid a_{1:H-1}, o_H) = \frac{1}{2^{H-1}}$$

for any $z \in \{u_1, u_2\}^{H-1}$ and $(a_{1:H-1}, o_H) \in [A]^{H-1} \times \{u_1, u_2\}$

Therefore, the observations at the first $H-1$ steps provide no information about the underlying transition. The only useful information is the last observation $o_H$ which tells us whether we end in good states or not.

3. The optimal policy is unique and is to execute the good action sequence $(a_1^\star, \ldots, a_{H-1}^\star)$ regardless of the obervations.

Based on the observations above, this is equivalent to a multi-arm bandits problem with $A^{H-1}$ arms. Therefore, we cannot do better than Brute-force search, which has sample complexity at least $\Omega(A^{H-1})$. $\qquad\square$

**Proposition 2.** *For any algorithm $\mathfrak{A}$, there exists an undercomplete POMDP ($S \leq O$) with $S$ and $O$ being small constants, such that algorithm $\mathfrak{A}$ requires at least $\Omega(A^{H-1})$ samples to ensure learning a $(1/4)$-optimal policy with probability at least $1/2$.*

*Proof.* We continue to use the POMDP constructed in Proposition 1 and slightly modify it by splitting $u_2$ into another 4 different observations $\{q_1, q_2, q_3, q_4\}$, so in the new POMDP ($O = 5 > S = 4$), we will observe a $q_i$ picked uniformly at random from $\{q_1, q_2, q_3, q_4\}$ when we are 'supposed' to observe $u_2$. It's easy to see the modification does not change its hardness. $\qquad\square$

# C Analysis of OMM-UCB

Throughout the proof, we use $\tau_h$ to denote a length-$h$ trajectory: $[a_h, o_h, \ldots, a_1, o_1] \in (\mathscr{A} \times \mathscr{O})^h$. Note that this definition is *different* from the one used in Section 5, where $\tau_h = [o_h, \ldots, a_1, o_1] \in \mathscr{O} \times (\mathscr{A} \times \mathscr{O})^{h-1}$ does not include the action $a_h$ at $h^{\text{th}}$ step. Besides, we define $\Gamma(\pi, h) = \{\tau_h = (a_h, o_h, \ldots, a_1, o_1) \mid \pi(a_h, \ldots, a_1 | o_h, \ldots, o_1) = 1\}$, which is also *different* from the definition in Section 5 wherer $a_h$ is not included.

Please refer to Appendix A for definitions of frequently used notations.

## C.1 Bounding the error in belief states

In this subsection, we will bound the error in (unnormalized) belief states, i.e., $\mathbf{b}(\tau_h; \theta) - \mathbf{b}(\tau_h; \hat{\theta})$ by the error in operators reweighed by the probability distribution of visited states.

We start by proving the following lemma that helps us decompose the error in belief states inductively.

**Lemma 7.** *Given a deterministic policy $\pi$ and two set of POMDP parameters $\hat{\theta} = (\hat{\mathbb{O}}, \hat{\mathbb{T}}, \hat{\mu}_1)$ and $\theta = (\mathbb{O}, \mathbb{T}, \mu_1)$, for all $h \geq 1$ and $\mathbf{X} \in \{\mathbf{I}_O, \hat{\mathbb{O}}_{h+1}^\dagger\}$, we have*

$$\sum_{\tau_h \in \Gamma(\pi,h)} \left\| \mathbf{X}\left(\mathbf{b}(\tau_h; \theta) - \mathbf{b}(\tau_h; \hat{\theta})\right) \right\|_1 \leq \sum_{\tau_{h-1} \in \Gamma(\pi,h-1)} \left\| \hat{\mathbb{O}}_h^\dagger \left(\mathbf{b}(\tau_{h-1}; \theta) - \mathbf{b}(\tau_{h-1}; \hat{\theta})\right) \right\|_1$$
$$+ \sum_{\tau_h \in \Gamma(\pi,h)} \left\| \mathbf{X}\left(\mathbf{B}_h(a_h, o_h; \hat{\theta}) - \mathbf{B}_h(a_h, o_h; \theta)\right) \mathbf{b}(\tau_{h-1}; \theta) \right\|_1.$$

*Proof.* By the definition of $\mathbf{b}(\tau_h; \theta)$ and $\mathbf{b}(\tau_h; \hat{\theta})$,

$$\sum_{\tau_h \in \Gamma(\pi,h)} \| \mathbf{X}\left(\mathbf{b}(\tau_h; \theta) - \mathbf{b}(\tau_h; \hat{\theta})\right) \|_1$$
$$= \sum_{\tau_h \in \Gamma(\pi,h)} \| \mathbf{X}\left(\mathbf{B}_h(a_h, o_h; \theta)\mathbf{b}(\tau_{h-1}; \theta) - \mathbf{B}_h(a_h, o_h; \hat{\theta})\mathbf{b}(\tau_{h-1}; \hat{\theta})\right) \|_1$$
$$\leq \sum_{\tau_h \in \Gamma(\pi,h)} \| \mathbf{X}\mathbf{B}_h(a_h, o_h; \hat{\theta}) \left(\mathbf{b}(\tau_{h-1}; \theta) - \mathbf{b}(\tau_{h-1}; \hat{\theta})\right) \|_1$$
$$+ \sum_{\tau_h \in \Gamma(\pi,h)} \| \mathbf{X}\left(\mathbf{B}_h(a_h, o_h; \hat{\theta}) - \mathbf{B}_h(a_h, o_h; \theta)\right) \mathbf{b}(\tau_{h-1}; \theta)\|_1.$$

The first term can be bounded as following,

$$\sum_{\tau_h \in \Gamma(\pi,h)} \| \mathbf{X}\mathbf{B}_h(a_h, o_h; \hat{\theta})(\mathbf{b}(\tau_{h-1}; \theta) - \mathbf{b}(\tau_{h-1}; \hat{\theta})) \|_1$$
$$= \sum_{\tau_h \in \Gamma(\pi,h)} \| \mathbf{X}\hat{\mathbb{O}}_{h+1}\hat{\mathbb{T}}_h(a_h)\text{diag}(\hat{\mathbb{O}}_h(o_h \mid \cdot))\hat{\mathbb{O}}_h^\dagger \left(\mathbf{b}(\tau_{h-1}; \theta) - \mathbf{b}(\tau_{h-1}; \hat{\theta})\right) \|_1$$
$$\leq \sum_{\tau_h \in \Gamma(\pi,h)} \sum_i \left\| \left(\mathbf{X}\hat{\mathbb{O}}_{h+1}\hat{\mathbb{T}}_h(a_h)\text{diag}(\hat{\mathbb{O}}_h(o_h \mid \cdot))\right)_i \right\|_1 \left| \left(\hat{\mathbb{O}}_h^\dagger \left(\mathbf{b}(\tau_{h-1}; \theta) - \mathbf{b}(\tau_{h-1}; \hat{\theta})\right)\right)_i \right|$$
$$= \sum_{\tau_h \in \Gamma(\pi,h)} \sum_i \left\| \left(\mathbf{X}\hat{\mathbb{O}}_{h+1}\hat{\mathbb{T}}_h(a_h)\right)_i \right\|_1 \hat{\mathbb{O}}_h(o_h \mid i) \left| \left(\hat{\mathbb{O}}_h^\dagger \left(\mathbf{b}(\tau_{h-1}; \theta) - \mathbf{b}(\tau_{h-1}; \hat{\theta})\right)\right)_i \right|$$
$$= \sum_{\tau_h \in \Gamma(\pi,h)} \sum_i \hat{\mathbb{O}}_h(o_h \mid i) \left| \left(\hat{\mathbb{O}}_h^\dagger \left(\mathbf{b}(\tau_{h-1}; \theta) - \mathbf{b}(\tau_{h-1}; \hat{\theta})\right)\right)_i \right|,$$

where the inequality is by triangle inequality, and the last identity follows from $\hat{\mathbb{T}}_h(a_h)$ (when $\mathbf{X} = \hat{\mathbb{O}}_{h+1}^\dagger$) and $\hat{\mathbb{O}}_{h+1}\hat{\mathbb{T}}_h(a_h)$ (when $\mathbf{X} = \mathbf{I}_O$) having columns with $\ell_1$-norm equal to 1.

As $\pi$ is deterministic, $a_h$ is unique given $\tau_{h-1}$ and $o_h$. Therefore,

$$\sum_{\tau_h \in \Gamma(\pi,h)} \sum_i \hat{\mathbb{O}}_h(o_h \mid i) \left| \left( \hat{\mathbb{O}}_h^\dagger \left( \mathbf{b}(\tau_{h-1};\theta) - \mathbf{b}(\tau_{h-1};\hat{\theta}) \right) \right)_i \right|$$

$$= \sum_{\tau_{h-1} \in \Gamma(\pi,h-1)} \sum_{o_h} \sum_i \hat{\mathbb{O}}_h(o_h \mid i) \left| \left( \hat{\mathbb{O}}_h^\dagger \left( \mathbf{b}(\tau_{h-1};\theta) - \mathbf{b}(\tau_{h-1};\hat{\theta}) \right) \right)_i \right|$$

$$= \sum_{\tau_{h-1} \in \Gamma(\pi,h-1)} \sum_i \sum_{o_h} \hat{\mathbb{O}}_h(o_h \mid i) \left| \left( \hat{\mathbb{O}}_h^\dagger \left( \mathbf{b}(\tau_{h-1};\theta) - \mathbf{b}(\tau_{h-1};\hat{\theta}) \right) \right)_i \right|$$

$$= \sum_{\tau_{h-1} \in \Gamma(\pi,h-1)} \sum_i \left| \left( \hat{\mathbb{O}}_h^\dagger \left( \mathbf{b}(\tau_{h-1};\theta) - \mathbf{b}(\tau_{h-1};\hat{\theta}) \right) \right)_i \right|$$

$$= \sum_{\tau_{h-1} \in \Gamma(\pi,h-1)} \left\| \hat{\mathbb{O}}_h^\dagger \left( \mathbf{b}(\tau_{h-1};\theta) - \mathbf{b}(\tau_{h-1};\hat{\theta}) \right) \right\|_1,$$

which completes the proof. $\qquad\square$

By applying Lemma 7 inductively, we can bound the error in belief states by the projection of errors in operators on preceding belief states.

**Lemma 8.** *Given a deterministic policy $\pi$ and two sets of undercomplete POMDP parameters $\theta = (\mathbb{O}, \mathbb{T}, \mu_1)$ and $\hat{\theta} = (\hat{\mathbb{O}}, \hat{\mathbb{T}}, \hat{\mu}_1)$ with $\sigma_{\min}(\hat{\mathbb{O}}) \geq \alpha$, for all $h \geq 1$, we have*

$$\sum_{\tau_h \in \Gamma(\pi,h)} \left\| \mathbf{b}(\tau_h;\theta) - \mathbf{b}(\tau_h;\hat{\theta}) \right\|_1$$

$$\leq \frac{\sqrt{S}}{\alpha} \sum_{j=1}^h \sum_{\tau_j \in \Gamma(\pi,j)} \left\| \left( \mathbf{B}_j(a_j,o_j;\hat{\theta}) - \mathbf{B}_j(a_j,o_j;\theta) \right) \mathbf{b}(\tau_{j-1};\theta) \right\|_1 + \frac{\sqrt{S}}{\alpha} \left\| \mathbf{b}_0(\theta) - \mathbf{b}_0(\hat{\theta}) \right\|_1.$$

*Proof.* Invoking Lemma 7 with $\mathbf{X} = \hat{\mathbb{O}}_{j+1}^\dagger$, we have

$$\sum_{\tau_j \in \Gamma(\pi,j)} \| \hat{\mathbb{O}}_{j+1}^\dagger \left( \mathbf{b}(\tau_j;\theta) - \mathbf{b}(\tau_j;\hat{\theta}) \right) \|_1 \leq \sum_{\tau_{j-1} \in \Gamma(\pi,j-1)} \left\| \hat{\mathbb{O}}_j^\dagger \left( \mathbf{b}(\tau_{j-1};\theta) - \mathbf{b}(\tau_{j-1};\hat{\theta}) \right) \right\|_1$$

$$+ \sum_{\tau_j \in \Gamma(\pi,j)} \| \hat{\mathbb{O}}_{j+1}^\dagger \left( \mathbf{B}_j(a_j,o_j;\hat{\theta}) - \mathbf{B}_j(a_j,o_j;\theta) \right) \mathbf{b}(\tau_{j-1};\theta) \|_1. \qquad (8)$$

Summing (8) over $j = 1, \ldots, h-1$, we obtain

$$\sum_{\tau_{h-1} \in \Gamma(\pi,h-1)} \| \hat{\mathbb{O}}_h^\dagger \left( \mathbf{b}(\tau_{h-1};\theta) - \mathbf{b}(\tau_{h-1};\hat{\theta}) \right) \|_1 \qquad\qquad\qquad (9)$$

$$\leq \sum_{j=1}^{h-1} \sum_{\tau_j \in \Gamma(\pi,j)} \left\| \hat{\mathbb{O}}_{j+1}^\dagger \left( \mathbf{B}_j(a_j,o_j;\hat{\theta}) - \mathbf{B}_j(a_j,o_j;\theta) \right) \mathbf{b}(\tau_{j-1};\theta) \right\|_1 + \left\| \hat{\mathbb{O}}_1^\dagger \left( \mathbf{b}_0(\theta) - \mathbf{b}_0(\hat{\theta}) \right) \right\|_1.$$

Again, invoking Lemma 7 with $\mathbf{X} = \mathbf{I}_O$ gives

$$\sum_{\tau_h \in \Gamma(\pi,h)} \| \mathbf{b}(\tau_h;\theta) - \mathbf{b}(\tau_h;\hat{\theta}) \|_1 \leq \sum_{\tau_{h-1} \in \Gamma(\pi,h-1)} \| \hat{\mathbb{O}}_h^\dagger ( \mathbf{b}(\tau_{h-1};\theta) - \mathbf{b}(\tau_{h-1};\hat{\theta}) ) \|_1$$

$$+ \sum_{\tau_h \in \Gamma(\pi,h)} \| \left( \mathbf{B}_h(a_h,o_h;\hat{\theta}) - \mathbf{B}_h(a_h,o_h;\theta) \right) \mathbf{b}(\tau_{h-1};\theta) \|_1. \qquad (10)$$

Plugging (9) into (10), and using the fact that $\| \hat{\mathbb{O}}_h^\dagger \|_{1 \to 1} \leq \sqrt{S} \| \hat{\mathbb{O}}_h^\dagger \|_2 \leq \frac{\sqrt{S}}{\alpha}$ complete the proof. $\quad\square$

The following lemma bounds the projection of any vector on belief states by its projection on the product of the observation matrix and the transition matrix, reweighed by the visitation probability of states.

**Lemma 9.** *For any deterministic policy $\pi$, fixed $a_{h+1} \in \mathscr{A}$, $\mathbf{u} \in \mathbb{R}^O$, and $h \geq 0$, we have*

$$\sum_{o_{h+1} \in \mathscr{O}} \sum_{\tau_h \in \Gamma(\pi,h)} \left| \mathbf{u}^\top \mathbf{b}([a_{h+1}, o_{h+1}, \tau_h]; \theta) \right| \leq \sum_{s=1}^{S} |\mathbf{u}^\top (\mathbb{O}_{h+2} \mathbb{T}_{h+1}(a_{h+1}))_s| \mathbb{P}_\theta^\pi(s_{h+1} = s).$$

*Proof.* By definition, for any $[a_{h+1}, o_{h+1}, \tau_h] \in \mathscr{A} \times \mathscr{O} \times \Gamma(\pi, h)$, we have

$$\mathbf{b}([a_{h+1}, o_{h+1}, \tau_h]; \theta) = \mathbb{O}_{h+2} \mathbb{T}_{h+1}(a_{h+1}) \mathbb{P}_\theta^\pi(s_{h+1} = \cdot, [o_{h+1}, \tau_h]),$$

where $\mathbb{P}_\theta^\pi(s_{h+1} = \cdot, [o_{h+1}, \tau_h])$ is an $s$-dimensional vector, whose $i^{\text{th}}$ entry is equal to the probability of observing $[o_{h+1}, \tau_h]$ and reaching state $i$ at step $h + 1$ when executing policy $\pi$ in POMDP($\theta$).

Therefore,

$$\sum_{\tau_h \in \Gamma(\pi,h)} \sum_{o_{h+1} \in \mathscr{O}} \left| \mathbf{u}^\top \mathbf{b}([a_{h+1}, o_{h+1}, \tau_h]; \theta) \right|$$

$$= \sum_{\tau_h \in \Gamma(\pi,h)} \sum_{o_{h+1} \in \mathscr{O}} \left| \mathbf{u}^\top \mathbb{O}_{h+2} \mathbb{T}_{h+1}(a_{h+1}) \mathbb{P}_\theta^\pi(s_{h+1} = \cdot, [o_{h+1}, \tau_h]) \right|$$

$$\leq \sum_{\tau_h \in \Gamma(\pi,h)} \sum_{o_{h+1} \in \mathscr{O}} \sum_{s=1}^{S} |\mathbf{u}^\top (\mathbb{O}_{h+2} \mathbb{T}_{h+1}(a_{h+1}))_s| \mathbb{P}_\theta^\pi(s_{h+1} = s, [o_{h+1}, \tau_h])$$

$$= \sum_{s=1}^{S} |\mathbf{u}^\top (\mathbb{O}_{h+2} \mathbb{T}_{h+1}(a_{h+1}))_s| \left( \sum_{\tau_h \in \Gamma(\pi,h)} \sum_{o_{h+1} \in \mathscr{O}} \mathbb{P}_\theta^\pi(s_{h+1} = s, [o_{h+1}, \tau_h]) \right)$$

$$= \sum_{s=1}^{S} |\mathbf{u}^\top (\mathbb{O}_{h+2} \mathbb{T}_{h+1}(a_{h+1}))_s| \mathbb{P}_\theta^\pi(s_{h+1} = s). \qquad \square$$

Combining Lemma 8 and Lemma 9, we obtain the target bound.

**Lemma 10.** *Given a deterministic policy $\pi$ and two sets of undercomplete POMDP parameters $\theta = (\mathbb{O}, \mathbb{T}, \mu_1)$ and $\hat{\theta} = (\hat{\mathbb{O}}, \hat{\mathbb{T}}, \hat{\mu}_1)$ with $\sigma_{\min}(\hat{\mathbb{O}}) \geq \alpha$, for all $h \geq 1$, we have*

$$\sum_{\tau_h \in \Gamma(\pi,h)} \|\mathbf{b}(\tau_h; \theta) - \mathbf{b}(\tau_h; \hat{\theta})\|_1$$

$$\leq \frac{\sqrt{S}}{\alpha} \left\| \mathbf{b}_0(\theta) - \mathbf{b}_0(\hat{\theta}) \right\|_1 + \frac{\sqrt{S}}{\alpha} \sum_{(a,o) \in \mathscr{A} \times \mathscr{O}} \left\| \left( \mathbf{B}_1(a, o; \hat{\theta}) - \mathbf{B}_1(a, o; \theta) \right) \mathbf{b}_0(\theta) \right\|_1$$

$$+ \frac{\sqrt{S}}{\alpha} \sum_{j=2}^{h} \sum_{(a,\tilde{a},o) \in \mathscr{A}^2 \times \mathscr{O}} \sum_{s=1}^{S} \left\| \left( \mathbf{B}_j(a, o; \hat{\theta}) - \mathbf{B}_j(a, o; \theta) \right) (\mathbb{O}_j \mathbb{T}_{j-1}(\tilde{a}))_s \right\|_1 \mathbb{P}_\theta^\pi(s_{j-1} = s).$$

*Proof.* By Lemma 8,

$$\sum_{\tau_h \in \Gamma(\pi,h)} \|\mathbf{b}(\tau_h; \theta) - \mathbf{b}(\tau_h; \hat{\theta})\|_1$$

$$\leq \frac{\sqrt{S}}{\alpha} \sum_{j=2}^{h} \sum_{\tau_j \in \Gamma(\pi,j)} \left\| \left( \mathbf{B}_j(a_j, o_j; \hat{\theta}) - \mathbf{B}_j(a_j, o_j; \theta) \right) \mathbf{b}(\tau_{j-1}; \theta) \right\|_1$$

$$+ \frac{\sqrt{S}}{\alpha} \sum_{\tau_1 \in \Gamma(\pi,1)} \left\| \left( \mathbf{B}_1(a_1, o_1; \hat{\theta}) - \mathbf{B}_1(a_1, o_1; \hat{\theta}) \right) \mathbf{b}_0(\theta) \right\|_1 + \frac{\sqrt{S}}{\alpha} \left\| \mathbf{b}_0(\theta) - \mathbf{b}_0(\hat{\theta}) \right\|_1. \quad (11)$$

**Bounding the first term:** note that $\Gamma(\pi, j) \subseteq \Gamma(\pi, j-2) \times (\mathscr{O} \times \mathscr{A})^2$, so we have

$$\sum_{\tau_j \in \Gamma(\pi,j)} \| \left( \mathbf{B}_j(a_j, o_j; \hat{\theta}) - \mathbf{B}_j(a_j, o_j; \theta) \right) \mathbf{b}(\tau_{j-1}; \theta) \|_1$$

$$\leq \sum_{\tau_{j-2} \in \Gamma(\pi,j-2)} \sum_{o_{j-1} \in \mathscr{O}} \sum_{a_{j-1} \in \mathscr{A}} \sum_{o_j \in \mathscr{O}} \sum_{a_j \in \mathscr{A}}$$

$$\| \left( \mathbf{B}_j(a_j, o_j; \hat{\theta}) - \mathbf{B}_j(a_j, o_j; \theta) \right) \mathbf{b}([a_{j-1}, o_{j-1}, \tau_{j-2}]; \theta) \|_1$$

$$= \sum_{(a_j, a_{j-1}, o_j) \in \mathscr{A}^2 \times \mathscr{O}}$$

$$\underbrace{\sum_{\tau_{j-2} \in \Gamma(\pi,j-2)} \sum_{o_{j-1} \in \mathscr{O}} \| \left( \mathbf{B}_j(a_j, o_j; \hat{\theta}) - \mathbf{B}_j(a_j, o_j; \theta) \right) \mathbf{b}([a_{j-1}, o_{j-1}, \tau_{j-2}]; \theta) \|_1}_{(\diamond)}. \qquad (12)$$

We can bound $(\diamond)$ by Lemma 9 and obtain,

$$\sum_{\tau_j \in \Gamma(\pi,j)} \| \left( \mathbf{B}_j(a_j, o_j; \hat{\theta}) - \mathbf{B}_j(a_j, o_j; \theta) \right) \mathbf{b}(\tau_{j-1}; \theta) \|_1$$

$$\leq \sum_{(a_j, a_{j-1}, o_j) \in \mathscr{A}^2 \times \mathscr{O}} \sum_{s=1}^{S} \| \left( \mathbf{B}_j(a_j, o_j; \hat{\theta}) - \mathbf{B}_j(a_j, o_j; \theta) \right) (\mathbb{O}_j \mathbb{T}_{j-1}(a_{j-1}))_s \|_1 \mathbb{P}_\theta^\pi(s_{j-1} = s)$$

$$= \sum_{(a, \tilde{a}, o) \in \mathscr{A}^2 \times \mathscr{O}} \sum_{s=1}^{S} \| \left( \mathbf{B}_j(a, o; \hat{\theta}) - \mathbf{B}_j(a, o; \theta) \right) (\mathbb{O}_j \mathbb{T}_{j-1}(\tilde{a}))_s \|_1 \mathbb{P}_\theta^\pi(s_{j-1} = s), \qquad (13)$$

where the identity only changes the notations $(a_j, a_{j-1}, o_j) \to (a, \tilde{a}, o)$ to make the expression cleaner.

**Bounding the second term:** note that $\Gamma(\pi, 1) \subseteq \mathscr{O} \times \mathscr{A}$, we have

$$\sum_{\tau_1 \in \Gamma(\pi,1)} \left\| \left( \mathbf{B}_1(a_1, o_1; \theta) - \mathbf{B}_1(a_1, o_1; \hat{\theta}) \right) \mathbf{b}_0(\theta) \right\|_1$$

$$\leq \sum_{(a,o) \in \mathscr{A} \times \mathscr{O}} \left\| \left( \mathbf{B}_1(a, o; \theta) - \mathbf{B}_1(a, o; \hat{\theta}) \right) \mathbf{b}_0(\theta) \right\|_1. \qquad (14)$$

Plugging (13) and (14) into (11) completes the proof. □

## C.2  A hammer for studying confidence sets

In this subsection, we develop a martingale concentration result, which forms the basis of analyzing confidence sets.

We start by giving the following basic fact about POMDP. The proof is just some basic algebraic calculation so we omit it here.

**Fact 11.** *In POMDP($\theta$), suppose $s_{h-1}$ is sampled from $\mu_{h-1}$, fix $a_{h-1} \equiv \tilde{a}$, and $a_h \equiv a$. Then the joint distribution of $(o_{h+1}, o_h, o_{h-1})$ is*

$$\mathbb{P}(o_{h+1} = \cdot, o_h = \cdot, o_{h-1} = \cdot) = (\mathbb{O}_{h+1} \mathbb{T}_h(a)) \otimes \mathbb{O}_h \otimes (\mathbb{O}_{h-1} \mathrm{diag}(\mu_{h-1}) \mathbb{T}_{h-1}(\tilde{a})^\top).$$

*By slicing the tensor, we can further obtain*

$$\begin{cases} \mathbb{P}(o_{h-1} = \cdot) = \mathbb{O}_{h-1} \mu_{h-1}, \\ \mathbb{P}(o_h = \cdot, o_{h-1} = \cdot) = \mathbb{O}_h \mathbb{T}_{h-1}(\tilde{a}) \mathrm{diag}(\mu_{h-1}) \mathbb{O}_{h-1}^\top, \\ \mathbb{P}(o_{h+1} = \cdot, o_h = o, o_{h-1} = \cdot) = \mathbb{O}_{h+1} \mathbb{T}_h(a) \mathrm{diag}(\mathbb{O}_h(o \mid \cdot)) \mathbb{T}_{h-1}(\tilde{a}) \mathrm{diag}(\mu_{h-1}) \mathbb{O}_{h-1}^\top. \end{cases}$$

A simple implication of Fact 11 is that if we execute policy $\pi$ from step 1 to step $h-2$, take $\tilde{a}$ and $a$ at step $h-1$ and $h$ respectively, then the joint distribution of $(o_{h+1}, o_h, o_{h-1})$ is the same as above except for replacing $\mu_{h-1}$ with $\mathbb{P}_\theta^\pi(s_{h-1} = \cdot)$.

Suppose we are given a set of sequential data $\{(o_{h+1}^{(t)}, o_h^{(t)}, o_{h-1}^{(t)})\}_{t=1}^N$ generated from POMDP($\theta$) in the following way: at time $t$, execute policy $\pi_t$ from step 1 to step $h-2$, take action $\tilde{a}$ at step $h-1$, and action $a$ at step $h$ respectively, and observe $(o_{h+1}^{(t)}, o_h^{(t)}, o_{h-1}^{(t)})$. Here, we allow the policy $\pi_t$ to be *adversarial*, in the sense that $\pi_t$ can be chosen based on $\{(\pi_i, o_{h+1}^{(i)}, o_h^{(i)}, o_{h-1}^{(i)})\}_{i=1}^{t-1}$. Define $\mu_{h-1}^{adv} = \frac{1}{N}\sum_{t=1}^N \mathbb{P}_\theta^{\pi_t}(s_{h-1} = \cdot)$. Based on Fact 11, we define the following probability vector, matrices and tensor,

$$
\begin{cases}
P_{h-1} = \mathbb{O}_{h-1}\mu_{h-1}^{adv}, \\
P_{h,h-1} = \mathbb{O}_h \mathbb{T}_{h-1}(\tilde{a})\mathrm{diag}(\mu_{h-1}^{adv})\mathbb{O}_{h-1}^\top, \\
P_{h+1,h,h-1} = (\mathbb{O}_{h+1}\mathbb{T}_h(a)) \otimes \mathbb{O}_h \otimes (\mathbb{O}_{h-1}\mathrm{diag}(\mu_{h-1}^{adv})\mathbb{T}_{h-1}(\tilde{a})^\top) \\
P_{h+1,o,h-1} = \mathbb{O}_{h+1}\mathbb{T}_h(a)\mathrm{diag}(\mathbb{O}_h(o \mid \cdot))\mathbb{T}_{h-1}(\tilde{a})\mathrm{diag}(\mu_{h-1}^{adv})\mathbb{O}_{h-1}^\top, \quad o \in \mathcal{O}.
\end{cases}
$$

Accordingly, we define their empirical estimates as below

$$
\begin{cases}
\hat{P}_{h-1} = \dfrac{1}{N}\sum_{t=1}^N \mathbf{e}_{o_{h-1}^{(t)}}, \\[2mm]
\hat{P}_{h,h-1} = \dfrac{1}{N}\sum_{t=1}^N \mathbf{e}_{o_h^{(t)}} \otimes \mathbf{e}_{o_{h-1}^{(t)}}, \\[2mm]
\hat{P}_{h+1,h,h-1} = \dfrac{1}{N}\sum_{t=1}^N \mathbf{e}_{o_{h+1}^{(t)}} \otimes \mathbf{e}_{o_h^{(t)}} \otimes \mathbf{e}_{o_{h-1}^{(t)}}, \\[2mm]
\hat{P}_{h+1,o,h-1} = \dfrac{1}{N}\sum_{t=1}^N \mathbf{e}_{o_{h+1}^{(t)}} \otimes \mathbf{e}_{o_{h-1}^{(t)}} \mathbf{1}(o_h^{(t)} = o), \quad o \in \mathcal{O}.
\end{cases}
$$

**Lemma 12.** *There exists an absolute constant $c_1$, s.t. the following concentration bound holds with probability at least $1 - \delta$*

$$
\max\left\{\|\hat{P}_{h+1,h,h-1} - P_{h+1,h,h-1}\|_F, \|\hat{P}_{h,h-1} - P_{h,h-1}\|_F, \right.
$$

$$
\left. \max_{o \in \mathcal{O}}\|\hat{P}_{h+1,o,h-1} - P_{h+1,o,h-1}\|_F, \|\hat{P}_{h-1} - P_{h-1}\|_2\right\} \leq c_1\sqrt{\frac{\log(ON/\delta)}{N}}.
$$

*Proof.* We start with proving that with probability at least $1 - \delta/2$,

$$
\|\hat{P}_{h+1,h,h-1} - P_{h+1,h,h-1}\|_F \leq c_1\sqrt{\frac{\log(ON/\delta)}{N}}.
$$

Let $\mathcal{F}_t$ be the $\sigma$-algebra generated by $\left\{\{\pi_i\}_{i=1}^{t+1}, \{(o_{h+1}^{(i)}, o_h^{(i)}, o_{h-1}^{(i)})\}_{i=1}^t\right\}$. $(\mathcal{F}_t)$ is a filtration. Define

$$
X_t = \mathbf{e}_{o_{h+1}^{(t)}} \otimes \mathbf{e}_{o_h^{(t)}} \otimes \mathbf{e}_{o_{h-1}^{(t)}} - (\mathbb{O}_{h+1}\mathbb{T}_h(a)) \otimes \mathbb{O}_h \otimes (\mathbb{O}_{h-1}\mathrm{diag}(\mathbb{P}_\theta^{\pi_t}(s_{h-1} = \cdot))\mathbb{T}_{h-1}(\tilde{a})^\top).
$$

We have $X_t \in \mathcal{F}_t$ and $\mathbb{E}[X_t \mid \mathcal{F}_{t-1}] = \mathbb{E}[X_t \mid \pi_t] = 0$, where the second identity follows from Fact 11. Moreover,

$$
\|X_t\|_F \leq \|X_t\|_1 \leq \|\mathbf{e}_{o_{h+1}^{(t)}} \otimes \mathbf{e}_{o_h^{(t)}} \otimes \mathbf{e}_{o_{h-1}^{(t)}}\|_1 +
$$

$$
\|(\mathbb{O}_{h+1}\mathbb{T}_h(a)) \otimes \mathbb{O}_h \otimes (\mathbb{O}_{h-1}\mathrm{diag}(\mathbb{P}_\theta^{\pi_t}(s_{h-1} = \cdot))\mathbb{T}_{h-1}(\tilde{a})^\top)\|_1 = 2, \tag{15}
$$

where $\|\cdot\|_1$ denotes the entry-wise $\ell_1$-norm of the tensor.

Now, we can bound $\|\hat{P}_{h+1,h,h-1} - P_{h+1,h,h-1}\|_F$ by writing $\hat{P}_{h+1,h,h-1} - P_{h+1,h,h-1}$ as the sum of a sequence of tensor-valued martingale difference, vectorizing the tensors, and applying the standard

vector-valued martingale concentration inequality (e.g. see Corollary 7 in [17]):

$$\|\hat{P}_{h+1,h,h-1} - P_{h+1,h,h-1}\|_F$$

$$= \|\frac{1}{N}\sum_{t=1}^{N}\left(e_{o_{h+1}^{(t)}} \otimes e_{o_h^{(t)}} \otimes e_{o_{h-1}^{(t)}} -\right.$$

$$(\mathbb{O}_{h+1}\mathbb{T}_h(a)) \otimes \mathbb{O}_h \otimes (\mathbb{O}_{h-1}\mathrm{diag}(\mathbb{P}_\theta^{\pi_t}(s_{h-1} = \cdot))\mathbb{T}_{h-1}(\tilde{a})^\top))\|_F$$

$$= \|\frac{1}{N}\sum_{t=1}^{N} X_t\|_F \leq \mathcal{O}\left(\sqrt{\frac{\log(ON/\delta)}{N}}\right),$$

with probability at least $1 - \delta/2$. We remark that when vectoring a tensor, its Frobenius norm will become the $\ell_2$-norm the vector. So the upper bound of the norm of the vectorized martingales directly follows from (15).

Similarly, we can show that with probability at least $1 - \delta/2$,

$$\|\hat{P}_{h,h-1} - P_{h,h-1}\|_F \leq \mathcal{O}\left(\sqrt{\frac{\log(ON/\delta)}{N}}\right) \quad \text{and} \quad \|\hat{P}_{h-1} - P_{h-1}\|_F \leq \mathcal{O}\left(\sqrt{\frac{\log(ON/\delta)}{N}}\right).$$

Using the fact $\|\hat{P}_{h+1,o,h-1} - P_{h+1,o,h-1}\|_F \leq \|\hat{P}_{h+1,h,h-1} - P_{h+1,h,h-1}\|_F$ completes the whole proof. $\square$

### C.3 Properties of confidence sets

For convenience of discussion, we divide the constraints in $\Theta_k$ into three categories as following

**Type-0 constraint:**
$$\|k \cdot \mathbf{b}_0(\hat{\theta}) - \mathbf{n}^k\|_2 \leq \beta_k\}$$

**Type-I constraint:**
$$\|\mathbf{B}_1(a, o; \hat{\theta})\mathbf{N}_1^k(a, \tilde{a}) - \mathbf{M}_1^k(o, a, \tilde{a})\|_F \leq \gamma_k,$$

where $\mathbf{M}_1^k$ and $\mathbf{N}_1^k$ are actually equivalent to $O$-dimensional counting vectors because there is no observation (or only a dummy observation) at step 0, which implies each of them has only one non-zero column. With slight abuse of notation, we use $\mathbf{M}_1^k$ and $\mathbf{N}_1^k$ to denote their non-zero columns in the following proof.

**Type-II constraint:** for $2 \leq h \leq H - 1$,
$$\|\mathbf{B}_h(a, o; \hat{\theta})\mathbf{N}_h^k(a, \tilde{a}) - \mathbf{M}_h^k(o, a, \tilde{a})\|_F \leq \gamma_k$$

Recalling the definition of $\mathbf{n}^k(\theta)$, $\mathbf{N}_h^k(a, \tilde{a})$ and $\mathbf{M}_h^k(o, a, \tilde{a})$ and applying Lemma 12, we get the following concentration results.

**Corollary 13.** *Let $\theta^\star = (\mathbb{T}, \mathbb{O}, \mu_1)$. By applying Lemma 12 directly, with probability at least $1 - \delta$, for all $k \in [K]$ and $(o, a, \tilde{a}) \in \mathscr{O} \times \mathscr{A}^2$, we have*

$$\begin{cases} \left\|\frac{1}{k}\mathbf{n}^k - \mathbb{O}_1\mu_1\right\|_2 \leq \mathcal{O}\left(\sqrt{\frac{\iota}{k}}\right), \\[2mm] \left\|\frac{1}{k}\mathbf{N}_1^k(a, \tilde{a}) - \mathbb{O}_1\mu_1\right\|_2 \leq \mathcal{O}\left(\sqrt{\frac{\iota}{k}}\right), \\[2mm] \left\|\frac{1}{k}\mathbf{M}_1^k(o, a, \tilde{a}) - \left(\mathbb{O}_2\mathbb{T}_1(\tilde{a})\mathrm{diag}(\mu_1)\mathbb{O}_1^\top\right)_o\right\|_2 \leq \mathcal{O}\left(\sqrt{\frac{\iota}{k}}\right), \\[2mm] \left\|\frac{1}{k}\mathbf{N}_h^k(a, \tilde{a}) - \underbrace{\mathbb{O}_h\mathbb{T}_{h-1}(\tilde{a})\mathrm{diag}(\mu_{h-1}^k)\mathbb{O}_{h-1}^\top}_{\mathbf{V}}\right\|_F \leq \mathcal{O}\left(\sqrt{\frac{\iota}{k}}\right), \\[2mm] \left\|\frac{1}{k}\mathbf{M}_h^k(o, a, \tilde{a}) - \underbrace{\mathbb{O}_{h+1}\mathbb{T}_h(a)\mathrm{diag}(\mathbb{O}_h(o \mid \cdot))\mathbb{T}_{h-1}(\tilde{a})\mathrm{diag}(\mu_{h-1}^k)\mathbb{O}_{h-1}^\top}_{\mathbf{W}}\right\|_F \leq \mathcal{O}\left(\sqrt{\frac{\iota}{k}}\right), \end{cases}$$

*where*

$$\iota = \log(KAOH/\delta) \quad \text{and} \quad \mu_{h-1}^k = \frac{1}{k}\sum_{t=1}^{k} \mathbb{P}_{\theta^\star}^{\pi_t}(s_{h-1} = \cdot) \quad 2 \le h \le H-1.$$

*Note that for all $k \in [K]$, $\mu_1^k = \mu_1$ independent of $\pi_1, \ldots, \pi_k$.*

Now, with Corollary 13, we can prove the true parameter $\theta^\star$ always lies in the confidence sets for $k \in [K]$ with high probability.

**Lemma 14.** *Denote by $\theta^\star = (\mathbb{T}, \mathbb{O}, \mu_1)$ the the ground truth parameters of the POMDP. With probability at least $1 - \delta$, we have $\theta^\star \in \Theta_k$ for all $k \in [K]$.*

*Proof.* By the definition of $\mathbf{b}_0(\theta^\star)$ and $\mathbf{B}_h(a, o; \theta^\star)$, we have

$$(*)\begin{cases} \mathbf{b}_0(\theta^\star) = \mathbb{O}_1\mu_1, \\ \left(\mathbb{O}_2\mathbb{T}_1(\tilde{a})\mathrm{diag}(\mu_1)\mathbb{O}_1^\top\right)_o = \mathbf{B}_1(\tilde{a}, o; \theta^\star)\mathbb{O}_1\mu_1, \\ \mathbf{W} = \mathbf{B}_h(a, o; \theta^\star) \cdot \mathbf{V}, \quad h \ge 2, \end{cases}$$

where $\mathbf{W}$ and $\mathbf{V}$ are shorthands defined in Corollary 13.

It's easy to see $(*)$ and Corollary 13 directly imply $\left\|\mathbf{n}^k - \mathbf{b}_0(\theta^\star)\right\|_2 \le \mathcal{O}\left(\sqrt{k\iota}\right)$ and thus $\theta^\star$ satisfies Type-0 constraint. For other constraints, we have

**Type-I constraint:**

$$\|\mathbf{M}_1^k(o, a, \tilde{a}) - \mathbf{B}_1(\tilde{a}, o; \theta^\star)\mathbf{N}_1^k(a, \tilde{a})\|_2$$
$$\le \|\mathbf{M}_1^k(o, a, \tilde{a}) - k\left(\mathbb{O}_2\mathbb{T}_1(\tilde{a})\mathrm{diag}(\mu_1)\mathbb{O}_1^\top\right)_o\|_2 + \|\mathbf{B}_1(\tilde{a}, o; \theta^\star)(k\mathbb{O}_1\mu_1 - \mathbf{N}_1^k(a, \tilde{a}))\|_2$$
$$\quad + k\|\left(\mathbb{O}_2\mathbb{T}_1(\tilde{a})\mathrm{diag}(\mu_1)\mathbb{O}_1^\top\right)_o - \mathbf{B}_1(\tilde{a}, o; \theta^\star)\mathbb{O}_1\mu_1\|_2$$
$$= \|\mathbf{M}_1^k(o, a, \tilde{a}) - k\left(\mathbb{O}_2\mathbb{T}_1(\tilde{a})\mathrm{diag}(\mu_1)\mathbb{O}_1^\top\right)_o\|_2 + \|\mathbf{B}_1(\tilde{a}, o; \theta^\star)(k\mathbb{O}_1\mu_1 - \mathbf{N}_1^k(a, \tilde{a}))\|_2$$
$$\le \|\mathbf{M}_1^k(o, a, \tilde{a}) - k\left(\mathbb{O}_2\mathbb{T}_1(\tilde{a})\mathrm{diag}(\mu_1)\mathbb{O}_1^\top\right)_o\|_2 + \|\mathbf{B}_1(\tilde{a}, o; \theta^\star)\|_2\|k\mathbb{O}_1\mu_1 - \mathbf{N}_1^k(a, \tilde{a})\|_2$$
$$\le \mathcal{O}\left(\frac{\sqrt{kS\iota}}{\alpha}\right)$$

where the identity follows from $(*)$, and the last inequality follows from Corollary 13 and

$$\|\mathbf{B}_h(a, o; \theta^\star)\|_2 = \|\mathbb{O}_{h+1}\mathbb{T}_h(a)\mathrm{diag}(\mathbb{O}_h(o|\cdot))\mathbb{O}_h^\dagger\|_2$$
$$\le \frac{1}{\alpha}\|\mathbb{O}_{h+1}\mathbb{T}_h(a)\mathrm{diag}(\mathbb{O}_h(o|\cdot))\|_2$$
$$\le \frac{\sqrt{S}}{\alpha}\|\mathbb{O}_{h+1}\mathbb{T}_h(a)\mathrm{diag}(\mathbb{O}_h(o|\cdot))\|_{1\to 1} \le \frac{\sqrt{S}}{\alpha}.$$

**Type-II constraint:** similarly, for $h \ge 2$, we have

$$\|\mathbf{B}_h(a, o; \theta^\star)\mathbf{N}_h^k(a, \tilde{a}) - \mathbf{M}_h^k(o, a, \tilde{a})\|_F$$
$$\le k\|\mathbf{B}_h(a, o; \theta^\star) \cdot \mathbf{V} - \mathbf{W}\|_F + \|\mathbf{B}_h(a, o; \theta^\star)(\mathbf{N}_h^k(a, \tilde{a}) - k\mathbf{V})\|_F + \|k\mathbf{W} - \mathbf{M}_h^k(o, a, \tilde{a})\|_F$$
$$= \|\mathbf{B}_h(a, o; \theta^\star)(\mathbf{N}_h^k(a, \tilde{a}) - k\mathbf{V})\|_F + \|k\mathbf{W} - \mathbf{M}_h^k(o, a, \tilde{a})\|_F$$
$$\le \|\mathbf{B}_h(a, o; \theta^\star)\|_2\|\mathbf{N}_h^k(a, \tilde{a}) - k\mathbf{V}\|_F + \|k\mathbf{W} - \mathbf{M}_h^k(o, a, \tilde{a})\|_F$$
$$\le \mathcal{O}\left(\frac{\sqrt{kS\iota}}{\alpha}\right),$$

Therefore, we conclude that $\theta^\star \in \Theta_k$ for all $k \in [K]$ with probability at least $1 - \delta$. $\qquad\square$

Furthermore, with Corollary 13, we can prove the following bound for operator error.

**Lemma 15.** *With probability at least $1-\delta$, for all $k \in [K]$, $\hat{\theta} = (\hat{\mathbb{O}}, \hat{\mathbb{T}}, \hat{\mu}_1) \in \Theta_{k+1}$ and $(o, a, \tilde{a}, h) \in \mathscr{O} \times \mathscr{A}^2 \times \{2, \ldots, H-1\}$, we have*

$$
\begin{cases}
\left\| \mathbf{b}_0(\theta^\star) - \mathbf{b}_0(\hat{\theta}) \right\|_2 \leq \mathcal{O}\left( \sqrt{\dfrac{\iota}{k}} \right), \\[2ex]
\left\| \left( \mathbf{B}_1(\tilde{a}, o; \hat{\theta}) - \mathbf{B}_1(\tilde{a}, o; \theta^\star) \right) \mathbf{b}_0(\theta^\star) \right\|_2 \leq \mathcal{O}\left( \sqrt{\dfrac{S\iota}{k\alpha^2}} \right), \\[2ex]
\displaystyle\sum_{s=1}^{S} \left\| \left( \mathbf{B}_h(a, o; \hat{\theta}) - \mathbf{B}_h(a, o; \theta^\star) \right) (\mathbb{O}_h \mathbb{T}_{h-1}(\tilde{a}))_s \right\|_1 \sum_{t=1}^{k} \mathbb{P}_{\theta^\star}^{\pi_t}(s_{h-1} = s) \leq \mathcal{O}\left( \sqrt{\dfrac{kS^2 O\iota}{\alpha^4}} \right),
\end{cases}
$$

*where $\iota = \log(KAOH/\delta)$.*

*Proof.* For readability, we copy the following set of identities from Lemma 14 here,

$$
(*) \begin{cases}
\mathbf{b}_0(\theta^\star) = \mathbb{O}_1 \mu_1, \\
\left( \mathbb{O}_2 \mathbb{T}_1(\tilde{a}) \mathrm{diag}(\mu_1) \mathbb{O}_1^\top \right)_o = \mathbf{B}_1(\tilde{a}, o; \theta^\star) \mathbb{O}_1 \mu_1, \\
\mathbf{W} = \mathbf{B}_h(a, o; \theta^\star) \cdot \mathbf{V}, \quad h \geq 2.
\end{cases}
$$

**Type-0 closeness:**

$$
\left\| \mathbf{b}_0(\theta^\star) - \mathbf{b}_0(\hat{\theta}) \right\|_2 \leq \left\| \frac{1}{k}\mathbf{n}^k - \mathbf{b}_0(\theta^\star) \right\|_2 + \left\| \frac{1}{k}\mathbf{n}^k - \mathbf{b}_0(\hat{\theta}) \right\|_2 \leq \mathcal{O}\left( \sqrt{\frac{\iota}{k}} \right),
$$

where the last inequality follows from $(*)$, Corollary 13 and $\hat{\theta} \in \Theta_{k+1}$.

**Type-I closeness:** similarly, we have

$$
\begin{aligned}
&\left\| \left( \mathbf{B}_1(\tilde{a}, o; \hat{\theta}) - \mathbf{B}_1(\tilde{a}, o; \theta^\star) \right) \mathbf{b}_0(\theta^\star) \right\|_2 \\
\leq &\left\| \left( \mathbb{O}_2 \mathbb{T}_1(\tilde{a}) \mathrm{diag}(\mu_1) \mathbb{O}_1^\top \right)_o - \mathbf{B}_1(\tilde{a}, o; \theta^\star) \mathbf{b}_0(\theta^\star) \right\|_2 \\
&+ \left\| \left( \mathbb{O}_2 \mathbb{T}_1(\tilde{a}) \mathrm{diag}(\mu_1) \mathbb{O}_1^\top \right)_o - \mathbf{B}_1(\tilde{a}, o; \hat{\theta}) \mathbf{b}_0(\theta^\star) \right\|_2 \\
= &\left\| \left( \mathbb{O}_2 \mathbb{T}_1(\tilde{a}) \mathrm{diag}(\mu_1) \mathbb{O}_1^\top \right)_o - \mathbf{B}_1(\tilde{a}, o; \hat{\theta}) \mathbf{b}_0(\theta^\star) \right\|_2 \\
\leq &\left\| \left( \mathbb{O}_2 \mathbb{T}_1(\tilde{a}) \mathrm{diag}(\mu_1) \mathbb{O}_1^\top \right)_o - \frac{1}{k}\mathbf{M}_1^k(o, a, \tilde{a}) \right\|_2 + \frac{1}{k}\left\| \mathbf{M}_1^k(o, a, \tilde{a}) - \mathbf{B}_1(\tilde{a}, o; \hat{\theta})\mathbf{N}_1^k(a, \tilde{a}) \right\|_2 \\
&+ \left\| \mathbf{B}_1(\tilde{a}, o; \hat{\theta}) \left( \frac{1}{k}\mathbf{N}_1^k(a, \tilde{a}) - \mathbf{b}_0(\theta^\star) \right) \right\|_2 \\
\leq &\left\| \left( \mathbb{O}_2 \mathbb{T}_1(\tilde{a}) \mathrm{diag}(\mu_1) \mathbb{O}_1^\top \right)_o - \frac{1}{k}\mathbf{M}_1^k(o, a, \tilde{a}) \right\|_2 + \frac{1}{k}\left\| \mathbf{M}_1^k(o, a, \tilde{a}) - \mathbf{B}_1(\tilde{a}, o; \hat{\theta})\mathbf{N}_1^k(a, \tilde{a}) \right\|_2 \\
&+ \left\| \mathbf{B}_1(\tilde{a}, o; \hat{\theta}) \right\|_2 \left\| \frac{1}{k}\mathbf{N}_1^k(a, \tilde{a}) - \mathbb{O}_1 \mu_1 \right\|_2 \\
\leq &\mathcal{O}\left( \sqrt{\frac{S\iota}{k\alpha^2}} \right),
\end{aligned}
$$

where the identity follows from $(*)$ and the last inequality follows from Corollary 13 and $\hat{\theta} \in \Theta_{k+1}$.

**Type-II closeness:** we continue to use the same proof strategy, for $h \geq 2$

$$\left\| \left( \mathbf{B}_h(a, o; \hat{\theta}) - \mathbf{B}_h(a, o; \theta^\star) \right) \mathbf{V} \right\|_F$$

$$\leq \|\mathbf{W} - \mathbf{B}_h(a, o; \theta^\star)\mathbf{V}\|_F + \|\frac{1}{k}\mathbf{M}_h^k(o, a, \tilde{a}) - \mathbf{W}\|_F$$

$$+ \frac{1}{k}\|\mathbf{B}_h(a, o; \hat{\theta})\mathbf{N}_h^k(a, \tilde{a}) - \mathbf{M}_h^k(o, a, \tilde{a})\|_F + \|\mathbf{B}_h(a, o; \hat{\theta}) \left( \mathbf{V} - \frac{1}{k}\mathbf{N}_h^k(a, \tilde{a}) \right)\|_F$$

$$= \|\frac{1}{k}\mathbf{M}_h^k(o, a, \tilde{a}) - \mathbf{W}\|_F + \frac{1}{k}\|\mathbf{B}_h(a, o; \hat{\theta})\mathbf{N}_h^k(a, \tilde{a}) - \mathbf{M}_h^k(o, a, \tilde{a})\|_F$$

$$+ \|\mathbf{B}_h(a, o; \hat{\theta}) \left( \mathbf{V} - \frac{1}{k}\mathbf{N}_h^k(a, \tilde{a}) \right)\|_F$$

$$\leq \mathcal{O}\left( \sqrt{\frac{S\iota}{k\alpha^2}} \right), \tag{16}$$

where the identity follows from $(*)$ and the last inequality follows from Corollary13 and $\hat{\theta} \in \Theta_{k+1}$. Recall $\mathbf{V} = \mathbb{O}_h \mathbb{T}_{h-1}(\tilde{a})\mathrm{diag}(\mu_{h-1}^k)\mathbb{O}_{h-1}^\top$ and utilize Assumption 1,

$$\left\| \left( \mathbf{B}_h(a, o; \hat{\theta}) - \mathbf{B}_h(a, o; \theta^\star) \right) \mathbf{V} \right\|_F$$

$$\geq \alpha \left\| \left( \mathbf{B}_h(a, o; \hat{\theta}) - \mathbf{B}_h(a, o; \theta^\star) \right) \mathbb{O}_h \mathbb{T}_{h-1}(\tilde{a})\mathrm{diag}(\mu_{h-1}^k) \right\|_F$$

$$\geq \frac{\alpha}{\sqrt{SO}} \left\| \left( \mathbf{B}_h(a, o; \hat{\theta}) - \mathbf{B}_h(a, o; \theta^\star) \right) \mathbb{O}_h \mathbb{T}_{h-1}(\tilde{a})\mathrm{diag}(\mu_{h-1}^k) \right\|_1$$

$$= \frac{\alpha}{k\sqrt{SO}} \sum_{s=1}^S \left\| \left( \mathbf{B}_h(a, o; \hat{\theta}) - \mathbf{B}_h(a, o; \theta^\star) \right) (\mathbb{O}_h \mathbb{T}_{h-1}(\tilde{a}))_s \right\|_1 \sum_{t=1}^k \mathbb{P}_{\theta^\star}^{\pi_t}(s_{h-1} = s).$$

Plugging it back into (16) completes the whole proof. $\square$

## C.4 Proof of Theorem 3

In order to utilize Lemma 15 to bound the operator error in Lemma 10, we need the following algebraic transformation. Its proof is postponed to Appendix E.

**Lemma 16.** *Let $z_k \in [0, C_z]$ and $w_k \in [0, C_w]$ for $k \in \mathbb{N}$. Define $S_k = \sum_{j=1}^k w_j$ and $S_0 = 0$. If $z_k S_{k-1} \leq C_z C_w C_0 \sqrt{k}$ for any $k \in [K]$, we have*

$$\sum_{k=1}^K z_k w_k \leq 2C_z C_w (C_0 + 1)\sqrt{K} \log(K).$$

*Moreover, there exists some hard case where we have a almost matching lower bound $O\left( C_z C_w C_0 \sqrt{K} \right)$.*

Now, we are ready to prove the main theorem based on Lemma 10, Lemma 15 and Lemma 16.

**Theorem 3.** *For any $\varepsilon \in (0, H]$, there exists $K_{\max} = \mathrm{poly}(H, S, A, O, \alpha^{-1})/\varepsilon^2$ and an absolute constant $c_1$, such that for any POMDP that satisfies Assumption 1, if we set hyperparameters $\beta_k = c_1\sqrt{k\log(KAOH)}$, $\gamma_k = \sqrt{S}\beta_k/\alpha$, and $K \geq K_{\max}$, then the output policy $\hat{\pi}$ of Algorithm 1 will be $\varepsilon$-optimal with probability at least $2/3$.*

*Proof.* There always exist an optimal deterministic policy $\pi^\star$ for the ground truth POMDP$(\theta^\star)$, i.e., $V^\star(\theta^\star) = V^{\pi^\star}(\theta^\star)$. WLOG, we can always choose the greedy policy $\pi_k$ to be deterministic, i.e., the probability to take any action given a history is either 0 or 1.

By Lemma 14, we have $\theta^\star \in \Theta_k$ for all $k \in [K]$ with probability at least $1 - \delta$. Recall that $(\pi_k, \theta_k) = \arg\max_{\pi, \theta \in \Theta_k} V^\pi(\theta)$, so with probability at least $1 - \delta$, we have

$$\sum_{k=1}^{K} \left( V^{\pi^\star}(\theta^\star) - V^{\pi_k}(\theta^\star) \right)$$

$$\leq \sum_{k=1}^{K} \left( V^{\pi_k}(\theta_k) - V^{\pi_k}(\theta^\star) \right)$$

$$\leq H \sum_{k=1}^{K} \sum_{[o_H, \tau_{H-1}] \in \mathscr{O} \times \Gamma(\pi_k, H-1)} \left\| \mathbb{P}_{\theta^\star}^{\pi_k}([o_H, \tau_{H-1}]) - \mathbb{P}_{\theta_k}^{\pi_k}([o_H, \tau_{H-1}]) \right\|_1$$

$$= H \sum_{k=1}^{K} \sum_{\tau_{H-1} \in \Gamma(\pi_k, H-1)} \left\| \mathbf{b}(\tau_{H-1}; \theta^\star) - \mathbf{b}(\tau_{H-1}; \theta_k) \right\|_1, \tag{17}$$

where the identity follows from Fact 18.

Applying Lemma 10, we have

$$\sum_{\tau_{H-1} \in \Gamma(\pi_k, H-1)} \left\| \mathbf{b}(\tau_{H-1}; \theta^\star) - \mathbf{b}(\tau_{H-1}; \theta_k) \right\|_1$$

$$\leq \underbrace{\frac{\sqrt{S}}{\alpha} \left\| \mathbf{b}_0(\theta^\star) - \mathbf{b}_0(\theta_k) \right\|_1 + \frac{\sqrt{S}}{\alpha} \sum_{(a,o) \in \mathscr{A} \times \mathscr{O}} \left\| \left( \mathbf{B}_1(a, o; \theta_k) - \mathbf{B}_1(a, o; \theta^\star) \right) \mathbf{b}_0(\theta^\star) \right\|_1}_{J_k}$$

$$+ \frac{\sqrt{S}}{\alpha} \sum_{h=2}^{H-1} \sum_{(a,\tilde{a},o) \in \mathscr{A}^2 \times \mathscr{O}} \sum_{s=1}^{S} \left\| \left( \mathbf{B}_h(a, o; \theta_k) - \mathbf{B}_h(a, o; \theta^\star) \right) \left( \mathbb{O}_h \mathbb{T}_{h-1}(\tilde{a}) \right)_s \right\|_1 \mathbb{P}_{\theta^\star}^{\pi_k}(s_{h-1} = s). \tag{18}$$

We can bound the first two terms by Lemma 15, and obtain that with probability at least $1 - \delta$,

$$H \sum_{k=1}^{K} J_k \leq \mathcal{O}\left( \frac{HSAO}{\alpha^2} \sqrt{K\iota} \right). \tag{19}$$

Plugging (18) and (19) into (17), we obtain

$$\sum_{k=1}^{K} \left( V^{\pi^\star}(\theta^\star) - V^{\pi_k}(\theta^\star) \right) \leq \mathcal{O}\left( \frac{HSAO}{\alpha^2} \sqrt{K\iota} \right) +$$

$$\frac{H^2 S^{1.5} A^2 O}{\alpha} \max_{s,o,a,\tilde{a},h} \sum_{k=1}^{K} \left\| \left( \mathbf{B}_h(a, o; \theta_k) - \mathbf{B}_h(a, o; \theta^\star) \right) \left( \mathbb{O}_h \mathbb{T}_{h-1}(\tilde{a}) \right)_s \right\|_1 \mathbb{P}_{\theta^\star}^{\pi_k}(s_{h-1} = s). \tag{20}$$

It remains to bound the second term.

By Lemma 15, with probability at least $1 - \delta$, for all $k \in [K]$, $\theta_k \in \Theta_k$ and $(s, o, a, \tilde{a}, h) \in \mathscr{S} \times \mathscr{O} \times \mathscr{A}^2 \times \{2, \ldots, H-1\}$, we have

$$\underbrace{\left\| \left( \mathbf{B}_h(a, o; \theta_k) - \mathbf{B}_h(a, o; \theta^\star) \right) \left( \mathbb{O}_h \mathbb{T}_{h-1}(\tilde{a}) \right)_s \right\|_1}_{z_k} \sum_{t=1}^{k-1} \underbrace{\mathbb{P}_{\theta^\star}^{\pi_t}(s_{h-1} = s)}_{w_t} \leq \mathcal{O}\left( \sqrt{\frac{kS^2 O \iota}{\alpha^4}} \right). \tag{21}$$

By simple calculation, we have $z_k \leq \sqrt{S}/\alpha$. Invoking Lemma 16 with (21), we obtain

$$\sum_{k=1}^{K} w_k z_k \leq \mathcal{O}\left( \frac{\sqrt{S^3 O \iota}}{\alpha^3} \sqrt{K} \log(K) \right). \tag{22}$$

Plugging (22) back into (20) gives

$$\sum_{k=1}^{K} \left( V^{\pi^\star}(\theta^\star) - V^{\pi_k}(\theta^\star) \right) \leq \mathcal{O}\left( \frac{H^2 S^3 A^2 O^{1.5} \sqrt{\iota}}{\alpha^4} \sqrt{K} \log(K) \right). \tag{23}$$

Finally, choosing

$$K_{\max} = \mathcal{O}\left( \frac{H^4 S^6 A^4 O^3 \log(HSAO/\varepsilon)}{\alpha^8 \varepsilon^2} \right),$$

and outputting a policy from $\{\pi_1, \ldots, \pi_K\}$ uniformly at random complete the proof. $\qquad\square$

# D   Learning POMDPs with Deterministic Transition

In this section, we introduce a computationally and statistically efficient algorithm for POMDPs with deterministic transition. A sketched proof is provided.

We comment that some previous works have studied POMDPs with deterministic transitions or deterministic emission process assuming the model is *known* (e.g. [4, 5, 6]); their results mainly focus on the planning aspect. In contrast, we assume *unknown* models which requires to learn the transition and emission process first. It is also worth mentioning that the (quasi)-deterministic POMDPs defined in these works are not exactly the same as ours. For example, the deterministic POMDPs in [6] refer to those with stochastic initial state but deterministic emission process, while we assume deterministic initial state but stochastic emission process. Therefore, their computational hardness results do not conflict with the efficient algorithm in this section.

---

**Algorithm 2** Learning POMDPs with Deterministic Transition

---

1: **initialize** $N = C \log(HSA/p)/(\min\{\epsilon/(\sqrt{O}H), \xi\})^2$, $n_h = \mathbf{1}(h = 1)$ for all $h \in [H]$.
2: **for** $h = 1, \ldots, H - 1$ **do**
3:     **for** $(s, a) \in [n_h] \times \mathscr{A}$ **do**
4:         Reset $z \leftarrow \mathbf{0}_{O \times 1}$ and $t \leftarrow n_{h+1} + 1$
5:         **for** $i \in [N]$ **do**
6:             execute policy $\pi_h(s)$ from step 1 to step $h - 1$, take action $a$ at $h^{\text{th}}$ step and observe $o_{h+1}$
7:             $z \leftarrow z + \frac{1}{N} e_{o_{h+1}}$
8:         **for** $s' \in [n_{h+1}]$ **do**
9:             **if** $\|\phi_{h+1,s'} - z\|_2 \leq 0.5\xi$ **then**
10:                $t \leftarrow s'$
11:         **if** $t = n_{h+1} + 1$ **then**
12:             $n_{h+1} \leftarrow n_{h+1} + 1$
13:             $\phi_{h+1,n_{h+1}} \leftarrow z$ and $\pi_{h+1}(n_{h+1}) \leftarrow a \circ \pi_h(s)$
14:         Set the $s^{\text{th}}$ column of $\hat{\mathbb{T}}_{h,a}$ to be $e_t$
15: **output** $\hat{\mu}_0 = e_1$ and $\left\{ n_h, \{\hat{\mathbb{T}}_{h,a}\}_{a \in \mathscr{A}} \text{ and } \{\phi_{h,i}\}_{i \in [n_h]} : h \in [H] \right\}$

---

**Theorem 4.** *For any $p \in (0, 1]$, there exists an algorithm such that for any deterministic transition POMDP satisfying Assumption 2, within $\mathcal{O}\left( H^2 SA \log(HSA/p)/(\min\{\varepsilon/(\sqrt{O}H), \xi\})^2 \right)$ samples and computations, the output policy of the algorithm is $\varepsilon$-optimal with probability at least $1 - p$.*

*Proof.* The algorithm works by inductively finding all the states we can reach at each step, utilizing the property of deterministic transition and good separation between different observation vectors. We sketch a proof based on induction below.

We say a state $s$ is $h$-step reachable if there exists a policy $\pi$ s.t. $\mathbb{P}^\pi(s_h = s) = 1$. In our algorithm, we use $n_h$ to denote the number of $h$-step reachable states. All the policies mentioned below is a sequence of fixed actions (independent of observations).

Suppose at step $h$, there are $n_h$ $h$-step reachable states and we can reach the $s^{\text{th}}$ one of them at the $h^{\text{th}}$ step by executing a *known* policy $\pi_h(s)$. Note that for every state $s'$ that is $(h+1)$-step reachable, there must exist some state $s$ and action $a$ s.t. $s$ is $h$-step reachable and $\mathbb{T}_h(s' \mid s, a) = 1$. Therefore, based on our induction assumption, we can reach all the $(h+1)$-step reachable states by executing all $a \circ \pi_h(s)$ for $(a, s) \in \mathscr{A} \times [n_h]$.

Now the problem is how to tell if we reach the same state by executing two different $a \circ \pi_h(s)$'s. The solution is to look at the distribution of $o_{h+1}$. Because the POMDP has deterministic transition, we always reach the same state when executing the same $a \circ \pi_h(s)$ and hence the distribution of $o_{h+1}$ is exactly the distribution of observation corresponding to that state. By Hoeffding's inequality, for each fixed $a \circ \pi_h(s)$, we can estimate the distribution of $o_{h+1}$ with $\ell_2$-error smaller than $\xi/8$ with high probability using $N \geq \tilde{\Omega}(1/\xi^2)$ samples. Since the observation distributions of two different states have $\ell_2$-separation no smaller than $\xi$, we can tell if two different $a \circ \pi_h(s)$'s reach the same

state by looking at the distance between their distributions of $o_{h+1}$. For those policies reaching the same state, we only need to keep one of them, so there are at most $S$ policies kept ($n_{h+1} \leq S$).

By repeating the argument inductively from $h = 1$ to $h = H$, we can recover the exact transition dynamics $\mathbb{T}_h(\cdot \mid s, a)$ and get an high-accuary estimate of $\mathbb{O}_h(\cdot \mid s)$ for every $h$-step reachable state $s$ and all $(h, a) \in [H] \times \mathscr{A}$ *up to permutation of states*. Since the POMDP has deterministic transition, we can easily find the optimal policy of the estimated model by dynamic programming.

The $\epsilon$-optimality simply follows from the fact that when $N \geq \tilde{\Omega}(H^2 O/\epsilon^2)$, we have the estimated distribution of observation for each state being $\mathcal{O}(\epsilon/H)$ accurate in $\ell_1$-distance for all reachable states. This implies that the optimal policy of the estimated model is at most $\mathcal{O}(\epsilon/H) \times H = \mathcal{O}(\epsilon)$ suboptimal. The overall sample complexity follows from our requirement $N \geq \max\{\tilde{\Omega}(H^2 O/\epsilon^2), \tilde{\Omega}(1/\xi^2)\}$, and the fact we need to run $N$ episodes for each $h \in [H], s \in \mathscr{S}, a \in \mathscr{A}$. $\qquad \square$

# E   Auxiliary Results

## E.1   Derivation of equation (2)

When conditioning on a fixed action sequence $\{a_{H-1}, \ldots, a_1\}$, a POMDP will reduce to a non-stationary HMM, whose transition matrix and observation matrix at $h^{\text{th}}$ step are $\mathbb{T}_h(a_h)$ and $\mathbb{O}_h$, respectively. So $\mathbb{P}(o_H, \ldots, o_1 | a_{H-1}, \ldots, a_1)$ is equal to the probability of observing $\{o_H, \ldots, o_1\}$ in this particular HMM. Using the basic properties of HMMs, we can easily express $\mathbb{P}(o_H, \ldots, o_1 | a_{H-1}, \ldots, a_1)$ in terms of the transition and observation matrices

$$\mathbb{O}_H(o_H|\cdot) \cdot [\mathbb{T}_{H-1}(a_{H-1})\text{diag}(\mathbb{O}_{H-1}(o_{H-1}|\cdot))] \cdots [\mathbb{T}_1(a_1)\text{diag}(\mathbb{O}_1(o_1|\cdot))] \cdot \mu_1.$$

Recall the definition of operators

$$\mathbf{B}_h(a, o) = \mathbb{O}_{h+1}\mathbb{T}_h(a)\text{diag}(\mathbb{O}_h(o|\cdot))\mathbb{O}_h^\dagger, \qquad \mathbf{b}_0 = \mathbb{O}_1\mu_1,$$

and $\mathbb{O}_h^\dagger \mathbb{O}_h = \mathbf{I}_S$, we conclude that

$$\mathbb{P}(o_H, \ldots, o_1 | a_{H-1}, \ldots, a_1) = \mathbf{e}_{o_H}^\top \cdot \mathbf{B}_{H-1}(a_{H-1}, o_{H-1}) \cdots \mathbf{B}_1(a_1, o_1) \cdot \mathbf{b}_0.$$

## E.2   Derivation of equation (6)

Note that $\pi$ is a deterministic policy and $\Gamma(\pi, H)$ is a set of all the observation and action sequences of length $H$ that could occur under policy $\pi$, i.e., for any $\tau_H = (o_H, \ldots, a_1, o_1) \in \Gamma(\pi, H)$, we have $\pi(a_{H-1} \ldots, a_1 \mid o_H, \ldots, o_1) = 1$, and $\pi(a'_{H-1} \ldots, a'_1 \mid o_H, \ldots, o_1) = 0$ for any action sequence $(a'_{H-1} \ldots, a'_1) \neq (a_{H-1} \ldots, a_1)$. Therefore, for $\tau_H \in \Gamma(\pi, H)$, we have:

$$
\begin{aligned}
\mathbb{P}_\theta^\pi(o_H, \ldots, o_1) &= \sum_{a'_{H-1} \in \mathscr{A}} \cdots \sum_{a'_1 \in \mathscr{A}} \mathbb{P}_\theta^\pi(o_H, a'_{H-1}, \ldots, a'_1, o_1) \\
&= \mathbb{P}_\theta^\pi(o_H, a_{H-1}, \ldots, a_1, o_1) \\
&= \left[ \prod_{h=1}^{H-1} \pi(a_h \mid o_h, a_{h-1}, \ldots, a_1, o_1) \right] \cdot \left[ \prod_{h=1}^{H} \mathbb{P}_\theta(o_h \mid a_{h-1}, o_{h-1} \ldots, a_1, o_1) \right] \\
&= \prod_{h=1}^{H} \mathbb{P}_\theta(o_h \mid a_{h-1}, o_{h-1}, \ldots, a_1, o_1) \\
&= \mathbb{P}_\theta(o_H, \ldots, o_1 | a_{H-1}, \ldots, a_1).
\end{aligned}
$$

## E.3   Boosting the success probability

We can run Algorithm 1 independently for $n = \mathcal{O}(\log(1/\delta))$ times and obtain $n$ policies. Each policy is independent of others and is $\varepsilon$-optimal with probability at least $2/3$. So with probability at least $1 - \delta/2$, at least one of them will be $\varepsilon$-optimal. In order to evaluate their performance, it suffices to run each policy for $\mathcal{O}(\log(n/\delta)H^2/\varepsilon^2)$ episodes and use the empirical average of the cumulative reward as an estimate. By standard concentration argument, with probability at least $1 - \delta/2$, the estimation error for each policy is smaller than $\varepsilon$. Therefore, if we pick the policy with the best empirical performance, then with probability at least $1 - \delta$, it is $3\varepsilon$-optimal. Rescaling $\varepsilon$ gives the desired accuracy. It is direct to see that the boosting procedure will only incur an additional $\text{polylog}(1/\delta)$ factor in the final sample complexity, and thus will not hurt the optimal dependence on $\varepsilon$.

## E.4   Basic facts about POMDPs and the operators

In this section, we provide some useful facts about POMDPs. Since their proofs are quite straightforward, we omit them here.

The following fact gives two linear equations the operators always satisfy. Its proof simply follows from the definition of the operators and Fact 11.

**Fact 17.** *In the same setting as Fact 11, suppose Assumption 1 holds, then we have*

$$\begin{cases} \mathbb{P}(o_h = \cdot, o_{h-1} = \cdot)\mathbf{e}_o = \mathbf{B}_h(\tilde{a}, o; \theta)\mathbb{P}(o_{h-1} = \cdot), \\ \mathbb{P}(o_{h+1} = \cdot, o_h = o, o_{h-1} = \cdot) = \mathbf{B}_h(a, o; \theta)\mathbb{P}(o_h = \cdot, o_{h-1} = \cdot). \end{cases}$$

The following fact relates (unnormalized) belief states to distributions of observable sequences. Its proof follows from simple computation using conditional probability formula and $\mathbb{O}_h^\dagger \mathbb{O}_h = \mathbf{I}_S$.

**Fact 18.** *For any POMDP($\theta$) satisfying Assumption 1, deterministic policy $\pi$ and $[o_h, \tau_{h-1}] \in \mathcal{O} \times \Gamma(\pi, h-1)$, we have*

$$\mathbf{e}_{o_h}^\top \mathbf{b}(\tau_{h-1}; \theta) = \mathbb{P}_\theta^\pi([o_h, \tau_{h-1}]),$$

*where $\mathbb{P}_\theta^\pi([o_h, \tau_{h-1}])$ is the probability of observing $[o_h, \tau_{h-1}]$ when executing policy $\pi$ in POMDP($\theta$).*

### E.5 Proof of Lemma 16

*Proof.* WLOG, assume $C_z = C_w = 1$. Let $n = \min\{k \in [K] : S_k \geq 1\}$. We have

$$\sum_{k=1}^{K} z_k w_k = \sum_{k=1}^{n} z_k w_k + \sum_{k=n+1}^{K} z_k w_k \leq \sum_{k=1}^{n} w_k + \sum_{k=n+1}^{K} z_k w_k$$

$$= S_n + \sum_{k=n+1}^{K} z_k w_k$$

$$\leq 2 + \sum_{k=n+1}^{K} z_k w_k.$$

It remains to bound the second term. Using the condition that $z_k S_{k-1} \leq C_0 \sqrt{k}$ for all $k \in [K]$, we have $z_k \leq \frac{C_0 \sqrt{k}}{S_{k-1}}$ for all $k \in [K]$ and $i \in [m]$. Therefore,

$$\sum_{k=n+1}^{K} z_k w_k \leq \sum_{k=n+1}^{K} C_0 \sqrt{k} \frac{w_k}{S_{k-1}}$$

$$\leq C_0 \sqrt{K} \sum_{k=n+1}^{K} \frac{w_k}{S_{k-1}}$$

$$\overset{(a)}{\leq} 2C_0 \sqrt{K} \sum_{k=n+1}^{K} \log(\frac{S_k}{S_{k-1}})$$

$$= 2C_0 \sqrt{K} \log(\frac{S_K}{S_n}) \leq 2C_0 \sqrt{K} \log(K),$$

where $(a)$ follows from $x \leq 2 \log(x+1)$ for $x \in [0, 1]$. $\qquad\square$

## Footnotes

[3]Note that this definition is *different* from the one used in Section 5, where $\tau_h = [o_h, \ldots, a_1, o_1] \in \mathscr{O} \times (\mathscr{A} \times \mathscr{O})^{h-1}$ does not include the action $a_h$ at $h^{\text{th}}$ step.

[4]WLOG, all the polices considered in this paper are *deterministic*. Also note that the trajectory in $\Gamma(\pi, h)$ contains $a_h$, which is *different* from the definition in Section 5