[Reviews · NeurIPS 2020]

Review 1

Summary and Contributions: The main contribution in the paper is to give an algorithm for learning near optimal policies in POMDPs with the optimal sample complexity, under minimal assumptions. Prior theoretical work on learning near optimal policies in POMDPs made strong reachability assumptions that made exploration simple, not capturing the exploration-exploitation tradeoff desirable in models of RL. This work learns near-optimal policies under the assumptions that (1) the POMDP is undercomplete and (2) its observation kernels satisfy a min singular value condition. Both are natural assumptions ((2) has appeared in the HMM literature before and POMDPs extend HMMs) and the paper gives two examples of POMDPs illustrating that these assumptions are statistically necessary. The algorithm is a UCB-type algorithm that uses ideas related to spectral methods for learning latent variable models in order to build confidence sets. In the case when the state-to-state transitions are deterministic, the paper gives a stronger algorithm that is also computationally efficient. UPDATE: Thank you to the authors for the clarifications and changes. My review remains that this paper should be a strong accept.

Strengths: POMDPs are an important class of MDPs in RL and little appears to be known theoretically about them. Furthermore, POMDPs are a technically difficult setup — as the optimal policy from an observation o_h depends on the entire observation-action history up to that point, the usual tools of the Bellman operator, Bellman rank and other recent advances in tabular MDPs do not apply. The main strength of the paper is that it gives a comprehensive first step towards understanding two key questions about POMDPs: (1) what is the optimal sample complexity/regret of learning near optimal policies, and (2) what conditions/assumptions are needed to learn? In particular, prior literature seems to have struggled with (2). From a technical perspective, the main novelty seems to be finding a way to use the undercompleteness to construct confidence sets (i.e. line 12 of Algorithm 1 and Lemma 6) that directly control regret (using the regret decompositions in equations (5) and (6)). This seems like an interesting technical contribution. The paper is also very well-written.

Weaknesses: While the main algorithm is statistically efficient, it is not computationally efficient and is likely impractical. Furthermore, the setting of deterministic state-to-state dynamics is fairly restrictive. However, I don’t think this should be viewed as a serious drawback — the setting of POMDPs is already technically difficult and restricting to the statistical question already produces a difficult problem which is comprehensively addressed in this paper.

Correctness: The proofs appear to be correct and are well-explained. This is a purely theoretical paper whose main algorithm is only statistically efficient (not computationally efficient) and thus has no empirical component.

Clarity: The paper is noticeably well-written and organized. This is a strength of the paper.

Relation to Prior Work: Section 1 does a very good job of discussing how this work fits into the surrounding literature. One comment is that the argument surrounding Lemma 6 and using undercompleteness seems to be related to the fact that the B matrices are low-rank (have rank <= S). It would be helpful if the authors addressed how the approach here compares to other rank-based arguments in the tabular MDP literature e.g. Bellman rank and the recent paper introducing FLAMBE by Agarwal et al. While it’s clear that Bellman rank cannot apply here directly (POMDPs are very far from tabular MDPs), does the argument here have any similarities to these other arguments?

Reproducibility: Yes

Additional Feedback: - Are there typos in the statements of Propositions 1 and 2: are undercomplete and overcomplete switched? - It would be helpful if Section 2.2 also discussed what implicit constraints equation (1) places on the B matrices e.g. that they are at most rank S.


Review 2

Summary and Contributions: The authors of this paper study the problem of PAC learning in a challenging problem of fixed horizon partially observable MDPs. In this paper, the authors first study lower bound with respect to the number of actions when their assumptions are not fulfilled. And then provide an algorithm with a PAC bound.

Strengths: The authors of this paper study the problem of PAC learning in a challenging problem of fixed horizon partially observable MDPs. In this paper, the authors first study lower bound with respect to the number of actions when their assumptions are not fulfilled. They show the lower bound is exponential in the number of actions, i.e., A^H where A is the number of actions, and H is the horizon of the problem. They do this by simply constructing a POMDP problem and show it is equivalent to a Bandit problem with A^H arms. This is the way the authors motivate their assumptions, which also has been adopted by prior works. The authors use the Observable Operator Model (OOMs) (details in Jaeger 2000) reformulation of POMDPs and build their algorithm based on optimism on OOM representation space. The proposed algorithm uses data from past episodes to estimates OOM parameters up to their confidence intervals. Then it chooses the most optimistic model in the constructed plausible set to come up with the policy deployed in the next episode. The authors then show that following this algorithm, one can come up with an epsilon-optimal policy with probability at least 2/3. This algorithm is statistically sound, but computationally, as argued by the authors, seems to be intractable. While the computation complexity is a caveat of this method, it might not be fair to ask the authors for a computationally efficient algorithm at the current stage of our understanding of POMDPs. If the proofs are correct, the results are interesting.

Weaknesses: A few comments that are needed to be addressed: 1) The first comment is about the presentation of the derivations. There are steps in the appendix, and also in the main text that are skipped. Some of them took me a while to rederive, some I couldn't spend more time to rederive. Some steps are also taken as granted in the main text. It is useful to elaborate on them more. For example, the derivation of B in eq1. While being known in the PSR and OOM literature, it is helpful to have the derivation in the paper. It can surely stay in the appendix—also similar issue with the eq2 and line 212. 2) I could not figure out the equality between lines 270 and 280. I will ask other reviewers to see whether they could follow that step. It would be great if the authors could refer me to its derivation. Since it is one of the core steps if this paper, understanding this part is one of the keys for me to advocate accepting this paper. 3) In lemma 12, the authors provide their concentration inequalities. Please include the steps, especially from line 492.5 and inequality in line 492.6. (Maybe you find it useful also to put the numerical value of c1.) 4) While the derivation of the Lemma12 sounds correct, I am not sure the way it is used is appropriate. The authors use o_{h-1}, a_{h-1}, o_{h}, a_{h}, o_{h+1} to estimate the P_{h+1,h,h-1}. Then they use o_{h}, a_{h}, o_{h+1}, a_{h+1}, o_{h+2} to estimate the P_{h+2,h+1,h}. Therefore, they reuse a portion of samples used to estimate P_{h+1,h,h-1}, in estimating P_{h+2,h+1,h} which results probably some undesirable correlation. I could not find a part of the derivation with deals with this. What am I missing here? 5) The final bound holds with probably of delta=2/3. Please leave a discussion in the paper on how to get the bound to hold for any arbitrary delta. 6) The dimension dependence in the final results: it is alright to have ploy(dimension) stated in the theorem, but I feel like the authors could manage to explicitly state the dependence of their final results in the problem dimension. It could help to compare the results with UCRL2 or REGAL. I am asking this since history-dependent policies basically deal with the MDP version of the POMDP (states as the whole history), and it would be great to see how much in the dimensions you save here. Based on the claim in this paper, the save is exponential ( because \mathcal{O}((OA)^H) is number states in the MDP version of the POMDP). But it would be great to have it quantified. I have read the appendix, it is nicely written, but I encourage the authors again to add the skipped steps. 7) A minor comment. The authors show that statistics of o_{h-1}, a_{h-1}, o_{h}, a_{h}, o_{h+1} suffice to learn the desired parameters and get a pac learning algorithm. Having this tuple for learning reminds me of this work https://arxiv.org/abs/1810.07900 that I came across recently, which uses the same sequence for some policy gradient approach. Any comments or discussion would be helpful. I am willing to advocate for accepting this paper; if the authors reply to the comment 2,3,4 in the rebuttal and address the rest of the comments in the final version. The results are great; I appreciate this study.

Correctness: I checked the claims, I had a bit of a hard time to verify them. I asked the authors for a few parts. If the authors verify those few steps, the rest sounded correct to me.

Clarity: Yes, it is well written and well-motivated. There are steps in the deviations and justifications that are skipped which makes it hard to follow. But generally, it is good. I strongly encourage the authors to add those steps in the paper, otherwise, in case of acceptance, there might be problems in approving the camera-ready version.

Relation to Prior Work: The relationships are discussed fairly ok, the authors could expand more on the prior works (more on MDP side) but I do not have a strong opinion.

Reproducibility: Yes

Additional Feedback: --------- I decided to raise the score I made to this paper. In case of acceptance, I'll make my final pass to see whether the explaination and derivation issues are address. Generally, a very good paper.


Review 3

Summary and Contributions: This paper considers the reinforcement learning (RL) problem of learning an epsilon-optimal policy for a finite-horizon partially observable Markov decision process (POMDP), while requiring that only a number of samples polynomial in the model parameters is used. The paper proves that this is possible for the class of POMDPs with more observations than states, where the observation probabilities at any state are not a mixture of the observation probabilities of any other states.

Strengths: This work significantly expands the class of POMDPs for which epsilon-optimal policies can be learned with a polynomial number of samples. In particular, no assumptions on the state transition process are required, unlike in some earlier work (Guo et al., Azizzadenesheli et al.). The assumptions the paper makes are also well-justified, it is shown that in their absence the results do not hold.

Weaknesses: While the theory in this paper is solid, as a POMDP practicioner I am uncertain about the practical impact of the results of this paper. In particular, Algorithm 1 instructs to solve a POMDP for a large parameter set (Line 4), which is impractical. The paper does provide results for POMDPs with deterministic state transitions, which are interesting, while arguably placing strong assumptions on the state transition process.

Correctness: The paper is technically correct. Overall, the paper lays out proofs in sufficient detail so as to enable checking. I checked the results in Appendices B and D.

Clarity: The paper is very lucid with a clear structure, and easy to follow. I enjoyed reading it. Typos: Line 231: "computationally" -> "computational" There is an extra period in the heading of Section 4 Appendix Line 565: should have T_h(s' \mid s, a) or T_h(s \mid s', a)?

Relation to Prior Work: Related work is covered in substantial depth. For the results on POMDPs with deterministic state transitions, it could be helpful to draw comparisons to complexity results for model-based planning with more restricted deterministic POMDPs, e.g., Bonet: "Deterministic POMDPs Revisited", UAI 2009.

Reproducibility: Yes

Additional Feedback: It would be helpful to have further comments and insight on the practical impact of the results. I am curious about the claim in the Appendix Lines 578-589: "Since the POMDP has deterministic transition, we can easily find the optimal policy of the estimated model by dynamic programming" -- isn't this still intractable since the state is partially observed? -- Additional comments after author response -- I thank the authors for their response that addresses all my questions. This paper makes a significant contribution and will further be improved by taking into account the suggestions from the reviewers.

[Author Response · NeurIPS 2020]

We would like to thank all reviewers for their valuable comments and time. Please see responses below:

**Reviewer 1.**

- Regarding the connections to Bellman rank and FLAMBE, since the algorithm style and underlying mechanisms of their papers are quite different from ours, to our best knowledge, we are not aware of direct connections based on our current analysis. However, our approach is model-based, which at a high level bears some similarity to the witness rank approach. We believe that a modified version of witness rank might be feasible for POMDPs, and a rank-based algorithm is also possible if combined with our current operator analysis. We agree this is an interesting future direction that is worth exploring.

- Proposition 1 shows that RL of POMDP is intractable in general if we do *not* assume undercompleteness. So the hard instance constructed in Proposition 1 is *overcomplete*. Proposition 2 shows that only assuming undercompleteness is also not enough. So the hard instance constructed there is *undercomplete*.

- Yes, the operator B defined in equation (1) has the property that its rank is at most S. We will clarify this.

**Reviewer 2.** Regarding the seven questions in the weakness section:

- Thanks for the suggestion about derivations. We will add more explanations about these derivations.

- For the equation between Lines 279 and 280, we note that $\pi$ is a *deterministic* policy and $\Gamma(\pi, H)$ is a set of all the observation and action sequences of length $H$ that could occur under policy $\pi$, i.e., for any $\tau_H = (o_H, \ldots, a_1, o_1) \in \Gamma(\pi, H)$, we have $\pi(a_{H-1} \ldots, a_1 \mid o_H, \ldots, o_1) = 1$, and $\pi(a'_{H-1} \ldots, a'_1 \mid o_H, \ldots, o_1) = 0$ for any action sequence $(a'_{H-1} \ldots, a'_1) \neq (a_{H-1} \ldots, a_1)$. Therefore, for $\tau_H \in \Gamma(\pi, H)$, we have:

$$\mathbb{P}_\theta^\pi(o_H, \ldots, o_1) = \sum_{a'_{H-1} \in \mathscr{A}} \cdots \sum_{a'_1 \in \mathscr{A}} \mathbb{P}_\theta^\pi(o_H, a'_{H-1}, \ldots, a'_1, o_1) = \mathbb{P}_\theta^\pi(o_H, a_{H-1}, \ldots, a_1, o_1)$$

$$= [\prod_{h=1}^{H-1} \pi(a_h \mid o_h, \ldots, a_1, o_1)] \cdot [\prod_{h=1}^{H} \mathbb{P}_\theta(o_h \mid a_{h-1}, \ldots, a_1, o_1)] = \prod_{h=1}^{H} \mathbb{P}_\theta(o_h \mid a_{h-1}, \ldots, a_1, o_1)$$

$$= \mathbb{P}_\theta(o_H, \ldots, o_1 | a_{H-1}, \ldots, a_1).$$

- The inequality in Line 492 follows from standard vector-valued martingale concentration (e.g. see Corollary 7 in "A Short Note on Concentration Inequalities for Random Vectors with SubGaussian Norm" by Jin et al.). Here, we vectorize the tensor, then the Frobenius norm becomes the $\ell_2$-norm of the vector. The upper bound on the Forbenius norm of the tensor is given at the beginning of the proof (Lines 490-491). The stated result is missing a $\log |O|$ factor. We will correct this and explain further in the final version.

- Please see Lines 7-11 in Algorithm 1, for *each* $(h, a, \tilde{a})$ triple, we *re-execute* the policy from the first step (i.e. start a new episode) to collect data $(o_{h-1}, o_h, o_{h+1})$ for $\mathbf{M}_h$ and $\mathbf{N}_h$. Therefore the samples for constructing $\mathbf{M}_h/\mathbf{N}_h$ and $\mathbf{M}_{h+1}/\mathbf{N}_{h+1}$ in Algorithm 1 are coming from completely different episodes. That is, other than the roll-in policy is the same, the samples are independent/have no undesirable correlation.

- Our result holds for any absolute constant probability $p < 1$. For general probably $1 - \delta$, directly applying our result will incur an additional $\text{poly}(1/\delta)$ factor in the sample complexity. However, one can easily improve this dependency to only $\text{polylog}(1/\delta)$ factor by computing $\log(1/\delta)$ independent policies that are near-optimal with only constant probability each, and pick the best one by evaluating each policy for $O(H^2/\epsilon^2)$ episodes.

- Yes, our savings of sample complexity is exponential compared to the naive $(OA)^H$. Our polynomial dependency is $\mathcal{O}(O^4 S^7 A^4 H^6/\alpha^4)$. We will provide this explicit dimension dependence in the final version.

- Thanks for pointing out this related work, we will add a discussion in the final version. The idea of using statistics $o_{h-1}, o_h, o_{h+1}$ for learning parameters stems from the earlier HMM literature [11]. Azizzadenesheli et al. 2020 is different from us in that it considers only Markovian policies and does not address exploration.

**Reviewer 3.**

- Thanks for pointing out these typos! Yes, Line 565 should be $T_h(s \mid s', a)$.

- Thanks for the reference to Blai Bonet's paper. The main difference from our paper is that we assume deterministic initial state but stochastic emission process, while they assume stochastic initial state but deterministic emission process. In addition, their result is on planning while assuming the model is known, while our result requires the learning of the model. We will add more discussion on the related works of deterministic POMDPs.

- For the claim in Lines 578-579, since we assume the initial state is fixed (see definition in Line 237), once we have learned the underlying transition matrices from the stochastic observations (Algorithm 2), we can directly identify the current hidden state by looking only at the sequence of actions taken.

[Meta-Review · NeurIPS 2020]

The reviewers reached a consensus that this paper deserves acceptance to neurips.